

# Morphology of bar-built estuaries: relation between planform shape and depth distribution

Jasper R.F.W. Leuven, Sanja Selaković, and Maarten G. Kleinhans

Faculty of Geosciences, Utrecht University, PO-box 80115, 3508 TC Utrecht, The Netherlands

**Correspondence:** Jasper R.F.W. Leuven (j.r.f.w.leuven@uu.nl)

**Abstract.** Fluvial-tidal transitions in estuaries are used as major shipping fairways and are characterised by complex bar and channel patterns with a large biodiversity. Habitat suitability assessment and study of interactions between morphology and ecology therefore require bathymetric data. While imagery offers data of planform estuary dimensions, only for a few natural estuaries bathymetries are available. Here we study the relation between along-channel planform geometry, obtained as the

outline from imagery, and hypsometry, which characterises the distribution of along-channel and cross-channel bed-levels. We fitted the original function of Strahler (1952) to bathymetric data along four natural estuaries. Comparison to planform estuary shape shows that hypsometry is concave at narrow sections with large channels, while complex bar morphology results in more convex hypsometry. We found a relation between hypsometric function shape and the degree to which the estuary width deviates from an ideal convergent estuary, which is calculated from river width and mouth width. This implies that

the occurring bed level distributions depend on inherited Holocene topography and lithology. Our new empirical function predicts hypsometry and along-channel variation in intertidal and subtidal width. Combination with the tidal amplitude allows an estimate of inundation duration. A validation of the results on available bathymetry shows that predictions of intertidal and subtidal area are accurate within a factor 2 for estuaries of different size and character. Locations with major human influence deviate from the general trends, because dredging, dumping, land reclamation and other engineering measures cause

local deviations from the expected bed-level distributions. The bathymetry predictor can be used to characterise and predict estuarine subtidal and intertidal morphology in data-poor environments.

## 1   Introduction

Estuaries develop as a result of dynamic interactions between hydrodynamical conditions, sediment supply, underlying geology
and ecological environment (Townend, 2012; de Haas et al., 2017). One model for the resulting morphology is that of the 'ideal estuary' that is hypothesised to have along-channel uniform tidal range, constant depth and current velocity, and a channel width that exponentially converges in landward direction such that the loss of tidal energy by friction is balanced (Savenije, 2006; Townend, 2012; Savenije, 2015; Dronkers, 2017). One would expect that in this case the along-channel variation in hypsometry



is also negligible. However, natural estuaries deviate from the ideal ones as result of varying degree of sediment supply, lack of time for adaptation and sea-level rise (Townend, 2012; de Haas et al., 2017) and locations wider than ideal are filled with tidal bars (Leuven et al., 2017) (Fig. 1). Differences in bed-level profiles between ideal and non-ideal estuaries are further enhanced by damming, dredging, dumping, land reclamation and other human engineering (e.g. O'Connor, 1987; Wang and

Winterwerp, 2001; Lesourd et al., 2001; Jeuken and Wang, 2010; Wang et al., 2015). All these natural deviations from the ideal estuary mean that there is no straightforward relation between the planform geometry of the estuary and the hypsometry or distribution of depths.

Hypsometry captures key elements of geomorphological features (Strahler, 1952; Boon and Byrne, 1981; Dieckmann et al., 1987; Kirby, 2000; Townend, 2008, 2010; de Vet et al., 2017) (Fig. 2). The hypsometric method was developed by Strahler

(1952) and Boon and Byrne (1981) to relate planform area of a basin to elevation. Later the results were used to predict the influence of basin morphology on the asymmetry of the horizontal and vertical tides, to predict flood- or ebb-dominance and maturity of an estuary (Boon and Byrne, 1981; Wang et al., 2002; Moore et al., 2009; Friedrichs, 2010) and to characterise the trend of saltmarsh development (Gardiner et al., 2011; Hu et al., 2015). Furthermore, hypsometry was used as a data reduction method to characterise shapes of tidal bars (de Vet et al., 2017). Hypsometry has also been used to describe dimensions

of channels and tidal flats in an idealised model (Townend, 2010). Here, a parabolic shape was prescribed for the low water channel, a linear profile for the intertidal low zone and convex profile for the intertidal high zone. While these profiles are valid for perfectly converging channels, it is unknow to which extent they are applicable to estuaries with irregular planforms and whether the currently assumed profiles are valid to assess flood- or ebb-dominance. Hypsometric profiles and derived inundation duration are also relevant indicators for habitat composition and future transitions from mudflat to saltmarsh (Tow-

nend, 2008). In order to predict and characterise morphology and assess habitat area, we need along-channel and cross-channel bed-level predictions for systems without measured bathymetry (Wolanski and Elliott, 2015).

For only a few natural estuaries bathymetry is available, which leaves many alluvial estuaries with irregular planforms from all around the world underinvestigated. However, many are visible in detail on satellite imagery, which raises the question whether there is a relation between planform geometry and depth distribution. Such a relation is known to exist in rivers in the

form of hydraulic geometry depending on bar pattern, and meander pool depths depending on planform channel curvature (e.g. Kleinhans and van den Berg, 2011; van de Lageweg et al., 2016). It therefore seems likely that such a relation between the horizontal and vertical dimensions exists for sandy estuaries, but this is not reported in literature. Morphological models can simulate 3D bed-levels with considerable accuracy (van der Wegen and Roelvink, 2008, 2012; van Maren et al., 2015; Braat et al., 2017), but these models are computationally intensive and need calibration and specification of initial and boundary con-

ditions. To study unmapped systems with limited information, it would be useful to be able to estimate bed-level distributions from planform geometry. Here we investigate this relation.

Previously, we showed in Leuven et al. (2017) how locations and widths of tidal bars can be predicted from the excess width, which is the local width of the estuary minus the ideal estuary width. The summed width of bars in each cross-section was found to approximate the excess width. This theory describes bars as discrete recognisable elements truncated at low water

level on what is essentially a continuous field of bed elevation that changes in along-channel direction (Leuven et al., 2016).





**Figure 1.** Bathymetry from (a) the Western Scheldt (NL), (b) Dovey (Wales), (c) Eems (NL) and (d) Columbia River estuary (USA). Source: (a,c) Rijkswaterstaat (NL), (b) Natural Resources Wales, (d) Lower Columbia Estuary Partnership.





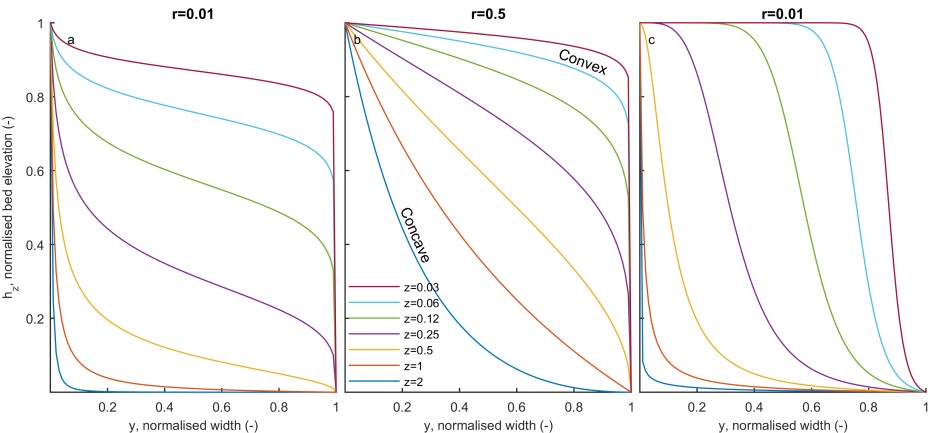

**Figure 2.** Hypsometric functions to describe morphological systems, modified from Strahler (1952). (a,b) Effect of $z$-values with $r$-value kept constant. (c) Inverted Strahler function (see Eq. 4).

However, predicting morphology in more detail requires predictions of along-channel and cross-channel bed elevations. While hypsometry can summarise bed elevation distribution as a cumulative profile, it is not clear whether the shape of the profile is predictable. Our hypothesis is that the along-channel variation in hypsometry depends on the degree to which an estuary deviates from its ideal shape. Therefore, we expect that locations with large excess width and thus large summed width of bars

have a more convex hypsometry (Fig. 3c,d). In case of ideal estuary width (almost) no bars (Fig. 3a,b), we expect concave hypsometry.

The aim of this manuscript is to investigate the relation between estuary planform outline and along-channel variation in hypsometry. To do so, first hypsometric curves are used to summarise the occurring bed elevations in a cumulative profile. Then, we use the original function of Strahler (1952) to fit the data obtained from bathymetry of four estuaries (Fig. 1). In the

10 results, we develop an empirical function to predict hypsometry. The quality and applications of the predictor are assessed in the discussion.

## 2 Methods

In this section, first the general form of a hypsometric curve is described. Second, the available datasets that were used for curve fitting are given. Last, the methodology to fit a hypsometric function to bathymetry in systems is presented.

### 2.1 General hypsometric curve

Strahler (1952) formulated the general hypsometric curve as:

$$h_z = \left[\frac{r}{r-1}\right]^z \left[\frac{1}{(1-r)y+r} - 1\right]^z \tag{1}$$





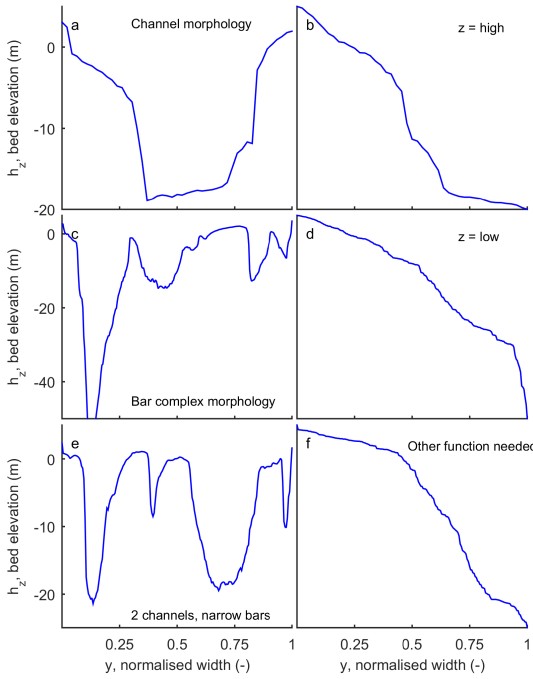

**Figure 3.** Example cross-sections and hypsometry, suggesting that channel-dominated morphology (a,b) generally results in concave hypsometric functions (high $z$-value in Fig. 2) while bar complexes (c,d) generally result in convex hypsometry (low $z$-value in Fig. 2). In case of narrow bars with a flat top and relatively steep transition from bar top to channel, the original hypsometry function by Strahler (1952) is less appropriate and the inverted function fits better.

in which $h_z$ is the value of the bed elevation, above which fraction $y$ of the width profile occurs. $r$ sets the slope of the curve at the inflection point in a range of 0.01-0.50, with sharper curves for lower values of $r$ (Fig. 2a,b). $z$ determines the concavity of the function in a range of 0.03-2, with lower values giving a more convex profile and higher values giving a more concave profile (Strahler, 1952) (Fig. 2). It is expected that $z$-values depend on excess width, because the fraction of the width occupied by bars becomes larger with excess width, resulting in a more convex hypsometric profile (Fig. 3c,d).

Excess width is defined as the local width minus the ideal width, which is given by:

$$W_{ideal}(x) = W_m \cdot e^{-x/L_W} \tag{2}$$



**Table 1.** Characteristics of estuaries used in this study.

| | $h_m$ (m) | $h_r$ (m) | $W_m$ (m) | $W_r$ (m) | $2a$ (m) | Area ($km^2$) | % intertidal | $Q_r$ ($m^3 s^{-1}$) |
|---|---|---|---|---|---|---|---|---|
| Western Scheldt | 25 | 15 | 4500 | 350 | 5 | 300 | 20 | 100 |
| Columbia river | 40 | 20 | 4000 | 800 | 2.5 | 900 | 30 | 7000 |
| Dovey | 10 | 2 | 450 | 50 | 3 | 12 | 75 | 30 |
| Eems | 25 | 8 | 3500 | 350 | 3.5 | 260 | 30 | 80 |

in which $x$ is the distance from the mouth, $W_m$ the width of the mouth, $e$ is $\approx 2.7$ and $L_W$ is the width convergence length (Davies and Woodroffe, 2010), which can be obtained conservatively from a fit on the width of the mouth and the landward river width (Leuven et al., 2017):

$$L_W = -s \frac{1}{\ln\left(\frac{W_s}{W_m}\right)} \tag{3}$$

in which $ln$ is the natural logarithm, $W_m$ is the local width measured at the mouth of the estuary, $W_s$ is the width measured at the landward side of the estuary and $s$ is the distance between these locations measured along the centreline. This practical method makes the convergence length somewhat sensitive to the selected position of the seaward and landward limit.

The landward limit was selected where the width ceases to converge on an image at the resolution of the full estuary scale. The seaward limit was selected as the location with the minimum width in case bedrock geology, human engineering or a

higher elevated spit confined the mouth, because in these cases the minimum width limits the inflow of tidal prism. In other cases, the mouth was chosen at the point where the first tidal flats were observed in the estuary or where the sandy beach ends at the mouth of the estuary. However, when the mouth is chosen at a location where sand bars are present, the ideal width will be overestimated and the width of intertidal area underestimated. It is therefore recommended to either chose the mouth at a location where bars are absent or subtract the width of bars from the measured with at the mouth to obtain the ideal width

profile.

## 2.2   Data availability and classification

Detailed bathymetries were available for four systems: the Western Scheldt (NL), Dovey (Wales), Eems (NL) and the Columbia River estuary (USA) (Fig. 1, Table 1). Data for the Western Scheldt and Eems were obtained from Rijkswaterstaat (NL), for the Dovey estuary from Natural Resources Wales and for the Columbia River estuary from the Lower Columbia Estuary Partner-

ship. Bed elevations were extracted from these bathymetries as follows. First, the estuary outline was digitised, excluding fully developed saltmarshes, and subsequently a centreline was determined within this polygon (following the approach of Leuven et al., 2017). Bed elevations were collected on equally spaced transects perpendicular to the centreline of the estuary. The data extracted at these transects was subsequently sorted by bed level value and made dimensionless to obtain hypsometric profiles (see Fig. 3 for examples).

We classified the transects by morphological characteristics and potential susceptibility to errors. The following morphological classes were used: mouth, bar junction, bar complex, narrow bar, pointbar, channel, pioneer marsh. The mouth is the



location where the estuary transitions into the sea. A bar junction is the most seaward or most landward tip of tidal bars. A bar complex is a location where a large bar is dissected by barb channels (Leuven et al., 2016) or multiple smaller bars are present. Narrow bar is used when the bars present were narrow along their entire length and often also relatively flat on their top. Pointbar is a bar in the inner bend of a large meander. Channel was assigned when bars were largely absent. Pioneer marsh

was assigned when aerial photographs or bathymetry gave visual indications of initial marsh formation, such as the presence of small tidal creeks and pioneering vegetation. Fully developed marsh is excluded from the outline.

The following classes were used to indicate possible errors: presence of harbours, major dredging locations, presence of a sand spit, presence of drainage channels for agriculture, constraints by hard layers, human engineering works. Either a locally deep channel or scour occurred at one of the sides of these transects or they lacked a natural transition from channel to estuary

bank, thus ending in their deepest part on one side of the transect. Major dredging locations have unnaturally deeper channels and shallower bars, resulting in a hypsometric shape that is relatively flat in the highest and lowest part and is steep in between (Fig. 3e,f). Furthermore, in a few cases side channels were perpendicular to the orientation of the main channel of the estuary. This resulted in transects being along-channel of these side channel, which biases transect data towards larger depth and creates a flat hypsometric profile at the depth of the side channel.

## 2.3   Data processing

Least-squares fits resulted in optimal values of $z$ and $r$ in Eq. 1 (Fig. 2) for each transect, using three different approaches. First, a regular least-squares curve fitting was used, which resulted in along-channel varying values for $z$, but an almost entirely constant along-channel value for $r$ of 0.5 (Fig. 4b,c, solid lines). In the second approach we set $r$ to a constant value of 0.5 and only fitted to obtain $z$ (Fig. 4c, dashed line). We found that the quality of the fit was the same, as indicated by the root-mean-

square error (RMSE) (Fig. 4d) and therefore apply this second approach in the remainder of this paper.

Locations where the RMSE was relatively large correspond to locations where major dredging occurred in the past century. This possibly resulted in a hypsometry characterised by a larger fraction of the width occupied by high tidal flats, a larger fraction of the width occupied by deep channels and a smaller fraction of the width occupied by the zone between channels and bars (Fig. 3e,f). Because the hypsometric curve at these locations deviated from the original Strahler function (Eq. 1), our third

approach was to apply a modified function to find optimal values for $z$ and $r$. To do so, the original formulation of Strahler (1952) was inverted to allow for hypsometries that describe steep transitions from bar top to channel bottom:

$$h_{z,inv} = \frac{\left[\frac{y^{1/z}(1-r)}{r} + 1\right]^{-1} - r}{1-r} \tag{4}$$

Applying this modified function resulted in better fits, but only at locations that were classified to be excluded because of possible errors. Therefore, results from this approach are not shown here and it is suggested to study the effect of dredging and

dumping on hypsometry in more detail in future studies.

In principle, both the bed elevation ($h_z$) and the width fraction $y$ in Eq. 1 are dimensionless. To compare the resulting predictions with measured values, the prediction needs to be dimensionalised. Values for $y$ are scaled with the local estuary





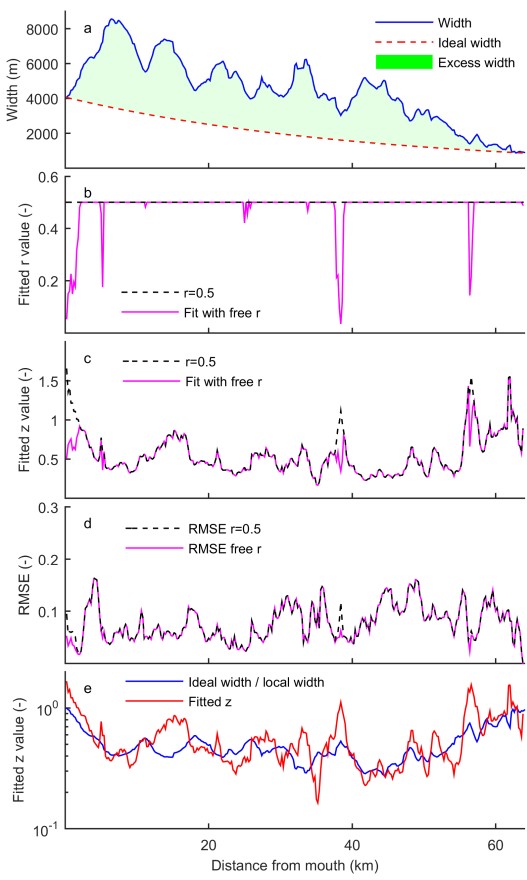

**Figure 4.** (a) Width along the Western Scheldt, with the maximum ideal converging width profile indicated. The green area is defined as the excess width (Leuven et al., 2017). For each along-channel transect of the estuary, the optimal fit of $z$ and $r$ in the Strahler (1952) function (Fig. 2) was determined. (b,c) Results for the Western Scheldt when both $z$ and $r$ are freely fitted as well as the results when $r$ is fixed to a constant value of 0.5. (d) Quality of fits remains about the same when $r$ is set to a fixed value of 0.5 as indicated by the root-mean-square error (RMSE). (e) Fitted $z$-values show similar trends as the ideal width divided by local width.



width. We test three options to scale $h_z$. The first option is to scale $h_z$ between the highest bed elevation and lowest bed elevation in the given cross-section, which is sensitive to the precise cut-off of the bathymetry. The second option is to scale $h_z$ between the local high water level (HWL) and the maximum estuary depth in that cross-section, which is sensitive to bathymetric information that is usually not available in unmapped estuaries.

The third option requires a prediction of depth at the upstream or downstream boundary. Width-averaged depth profiles along estuaries are often (near-) linear (Savenije, 2015; Leuven et al., 2017). Therefore, only the channel depth at the mouth of the estuary and at the upstream river have to be estimated and subsequently a linear regression can be made. Channel depth at upstream river ($h_r$) is estimated with hydraulic geometry relations (e.g. Leopold and Maddock Jr, 1953; Hey and Thorne, 1986): $h_r = 0.12 W_r^{0.78}$. The depth at the mouth is estimated from relations between tidal prism and cross-sectional area (e.g.

O'Brien, 1969; Eysink, 1990; Friedrichs, 1995; Lanzoni and D'Alpaos, 2015; Gisen and Savenije, 2015; Leuven et al., 2017). Here we used:

$$h_m = \frac{0.13 \cdot 10^{-3} P}{W_m} \tag{5}$$

in which $P$ is the tidal prism, which can be estimated by multiplying the estuary surface area with the tidal range (Leuven et al., 2017). Locally, the maximum depth may be deeper or lower than predicted, due to the presence of resistant layers in the

subsurface or where banks are fixed or protected. While this may affect the accuracy of the locally predicted maximum channel depth, it has a minor effect on the calculations of sub- and intertidal area. Moreover, the upper limit for dimensionalisation is chosen as the high water line, which implies that supratidal area is not included in the predictions. We will show results with all methods, but only the third can be applied when information about depth is entirely lacking.

Statistical analyses in the remainder of paper were approached as follows. In linear regressions, we minimised residuals

in both the x- and y-directions. This results in regressions that are more robust than when residuals in only one direction are minimised. In case regressions are plotted, the legend will specify the multiplication factor that the confidence limits plot above or below the trend. $R^2$ values are given to indicate the variance around the regression. In cases where the quality of correlation of two along-channel profiles is assessed, we used the Pearson product-moment correlation coefficient (r).

## 3 Results

We found a strong relation between along-channel variation in hypsometry and the degree to which an estuary deviates from its ideal shape. Below, we will first show how the hypsometry of typical channel deviates from that of bar complexes. Subsequently, the data is presented per system, classified on their morphology and potential for errors due to human interference, method and other causes. Then we combine all data to derive an empirical formulation to predict hypsometry. Last, we apply this formulation to predict the along-channel variation in intertidal and subtidal width and validate the results with measure-

ments from bathymetry.

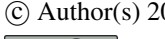



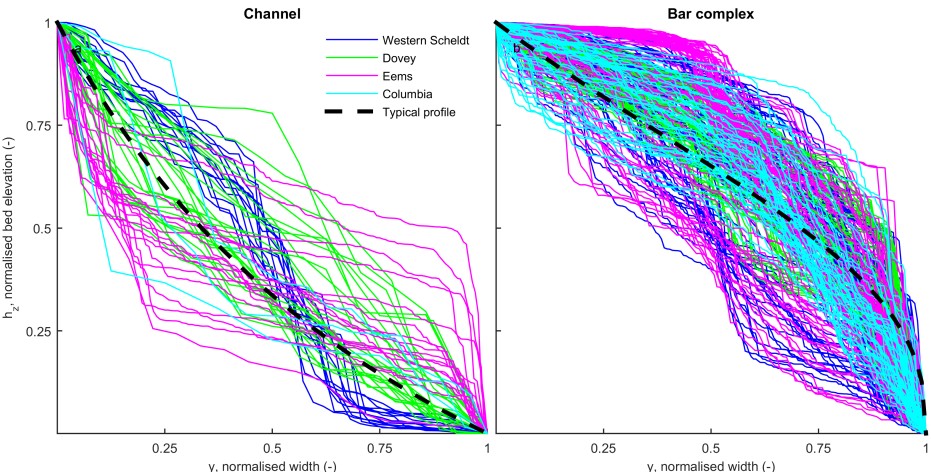

**Figure 5.** Hypsometric curves as extracted from bathymetry in the Western Scheldt, Dovey, Eems and Columbia River. Profiles were classified as channel when sand bars were (mostly) absent and as bar complex when one larger bar disected by barb channels or multiple smaller bars were present. Channel-dominated morphology generally results in concave hypsometric profiles and bar complexes in convex profiles.

## 3.1 Relation between morphology and hypsometry

As hypothesised, it is indeed observed that channel-dominated morphology results in more concave hypsometry profiles (high $z$-value), while bar complex morphology results in more convex hypsometry (low $z$-value) (Fig. 5). Values for $z$ in Eq. 1 range from 0.83 to 1.14 for channels, with an average value of 1.0. In contrast, $z$ ranges from 0.36 to 0.41 for bar complexes, with an average of 0.39.

Clustering of morphological classes strongly suggest a relation between hypsometry and planform estuary shape (Fig. 6). Mouth and channel-dominated morphology typically plot at the right-hand side of the plots in Fig. 6a,c,e,g, thus being locations close to ideal width. In case of the Western Scheldt this results in the highest values of $z$. In the case of the Dovey and Columbia River, the mouth region was influenced by respectively a spit and human engineering, which resulted in the formation of tidal flats on the side and thus lead to a lower $z$-value.

Bar complexes occur at the other end of the spectrum; these locations are generally much wider than the ideal shape and are characterised with hypsometries with a $z$-value well below 1. Bar junctions as well as narrow bars are generally found at the transition from channel-dominated morphology to bar complex morphology and therefore also occur between these types in the plots. The point bar in the Western Scheldt (the *Plaat van Ossenisse*) shows hypsometry comparable to bar complexes (Fig. 6a), which reflects the complex history of formation by multiple bar amalgamations. Also the locations in the Columbia River where pioneer marsh is present, show the same trend as the locations where unvegetated bar complexes occur (Fig. 6g).

In a few cases, the transects used to extract bathymetry were not perpendicular to the main channel of the estuary. For example, landward and seaward of the pointbar in the Western Scheldt (*Plaat van Ossenisse*) transects were inclined, covering a larger part of the channel than perpendicular transects, resulting in higher $z$-values as a consequence of the apparent channel-



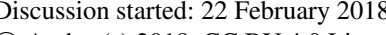

**Figure 6.** Results of hypsometry fitting, where clustering indicates a relation to planform geometry. (a,c,e,g) $z$-values were fitted for cross-channel transects in bathymetry of four estuaries and plotted by morphological classification. (b,d,f,h) Regressions for $z$-value as a function of ideal width divided by local width. Data points that were influenced by human interference, bedrock geology or errors in methodology or data indicated in red in panels a,c,e,g were excluded. Confidence limits are plotted at two standard deviations above and below the regression and their multiplication factor compared to the trend is given in the legend.



dominated morphology. Immediately landward of the spit in the Dovey, transects are almost parallel to the shallow side-channel. Fitting hypsometry at these locations resulted in relatively low $z$-values, because it is a relatively shallow side-channel.

For the Western Scheldt and Eems it is known at which locations major dredging and dumping takes place (e.g. Swinkels et al., 2009; Jeuken and Wang, 2010; Bolle et al., 2010; Dam et al., 2015; Plancke and Vos, 2016). Even though the resulting
$z$-values at these locations do not cause major outliers, the quality of fits are typically lower and the inverted Strahler function (Eq.4) fitted better. These points were therefore excluded from further analysis.

The filtered data shows quasi-cyclicity in along-channel hypsometry (Fig. 7). In general, the width at the mouth of the estuary and at the upstream estuary is close to ideal and the hypsometry is concave. The part in between is characterised by variations in the local width and therefore gradual increases and decreases in the ratio between local width and ideal width. In
some cases, quasi-cyclic loops are visible (e.g. Fig. 7c) caused by the asymmetry in bar complexes. In other cases, the points show more zigzag or clustered patterns, which indicate minor variation on the bar complexes or scatter in the fit applied to the bathymetry.

## 3.2    Hypsometry predictor

The relations between excess width and hypsometric function are similar for all estuaries, which suggest that a universal
function is of value. Combining all the filtered data resulted in a regression between the extent to which an estuary deviates from the ideal shape and the predicted $z$-value in the hypsometry formulation (Fig. 8). Data from Columbia River, Eems in 1985, Western Scheldt in 2013 and Dovey were used to obtain this relation. Other data are shown in Fig. 8 but not used in the regression.

These results mean that we found a predictive function for hypsometry, where $r$ is set to a constant value of 0.5 and the $z(x)$
is calculated as:

$$z(x) = 1.4 \left[ \frac{W_{ideal}(x)}{W(x)} \right]^{1.2} \tag{6}$$

in which $W_{ideal}(x)$ is the ideal estuary width (Eq. 2) and $W(x)$ is the measured local width. The confidence limits of the regression plot a factor 1.9 higher and lower than the regression, which indicates that the $z$-value can be predicted within a factor 2 (see Fig. 9 for an example of prediction with uncertainty). While not used in the regression, hypsometry from
bathymetry in other years show similar trends and scatter as the data used in the regression.

The predictor (Eq. 6) was applied to the Columbia River estuary and Western Scheldt to check the quality of the resulting along-channel predictions of intertidal high, intertidal low and subtidal width (Fig. 10). These zones can be derived after dimensionalising hypsometry and imposing a tidal range (Fig. 9b). For almost the entire along-channel profile, the predictions are within a factor 2 of the measured value (Fig. 11) and best agreement was obtained when the hypsometry was dimensionalised
between the minimum and maximum measured bed level for each transect (Fig. 10).





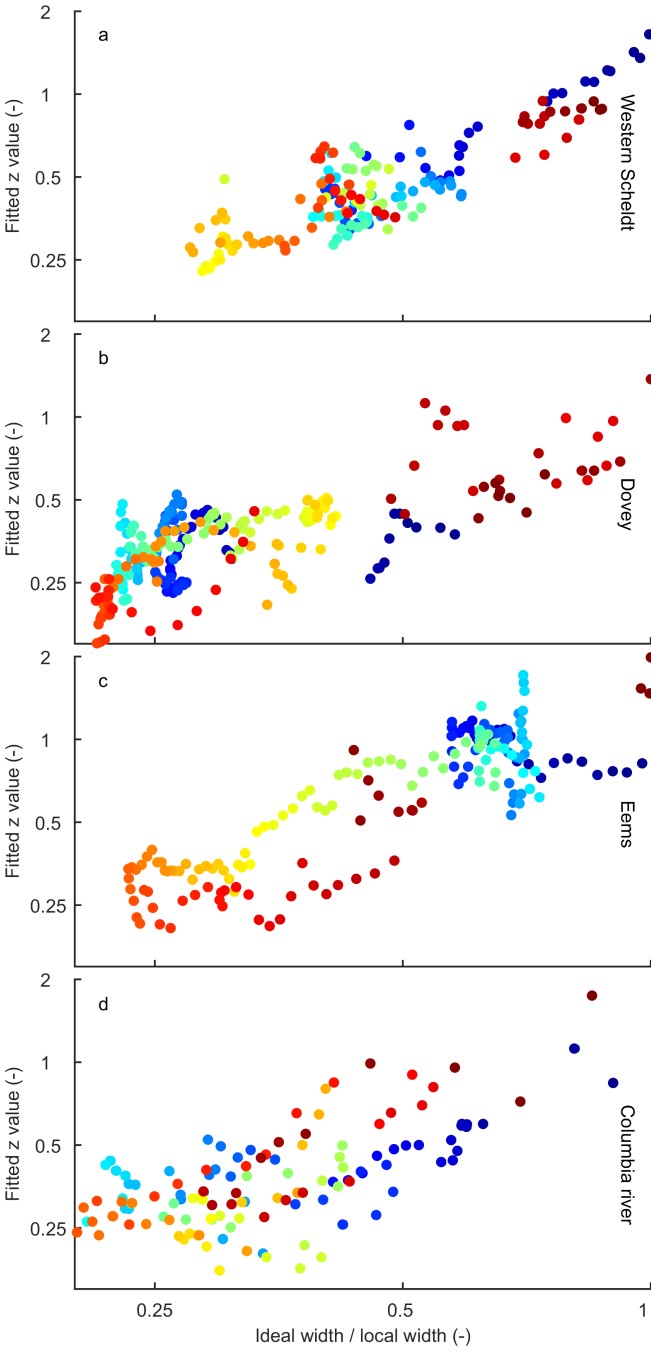

**Figure 7.** Fitted z-values as a function of deviation from the ideal width. Colours indicate the location along the estuary, with dark blue colours at the mouth transitioning into dark red colours at the landward end. Some zones show scatter in fitted $z$-values, while some other zones (e.g. green to orange to red in c) show quasi-periodic behaviour.





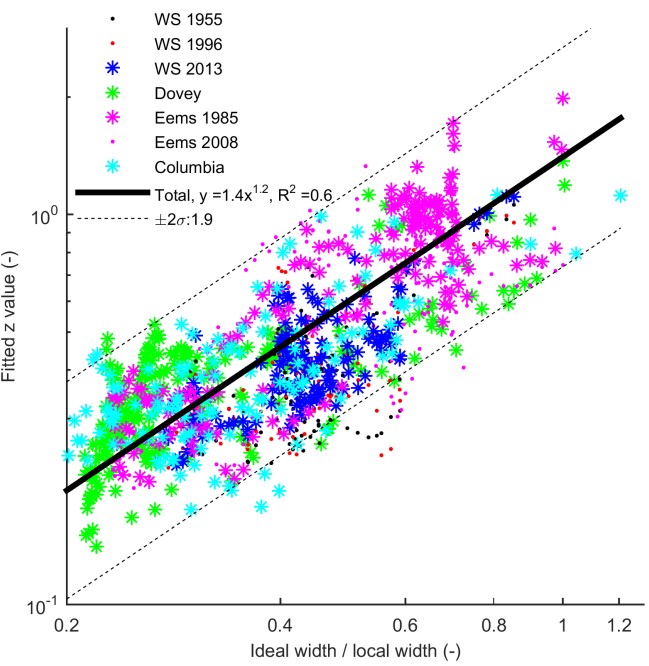

**Figure 8.** Fitted $z$-values of filtered data increase with the fraction of ideal width and local width, which indicate that hypsometric shapes become progressively more concave when the local width approaches the ideal width and become more convex when the local width becomes larger than the ideal width (i.e. the excess width increases). The data shown as asterisks is used for the regression. Confidence limits are plotted at two standard deviations above and below the regression and their multiplication factor compared to the trend is given in the legend.

**Table 2.** Percentage of points predicted within a factor 2 from the measured value.

| Estuary | % for subtidal | % for intertidal |
|---|---|---|
| Western Scheldt | 100 | 84 |
| Columbia river | 90 | 79 |
| Dovey | 54 | 71 |
| Eems | 91 | 59 |

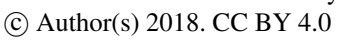



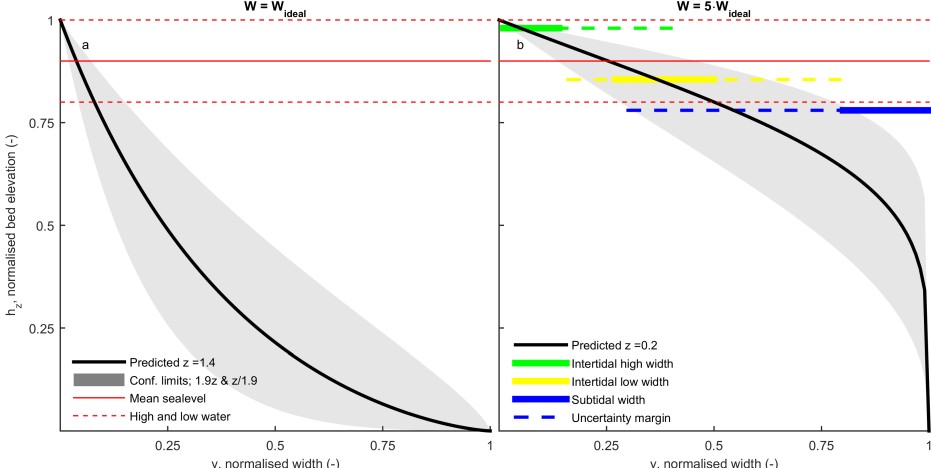

**Figure 9.** Illustration of uncertainty of predicted hypsometry from Eq. 6 with uncertainty margins (Fig. 8). Resulting prediction for hypothethical location where (a) local width is equal to the ideal width and (b) local width is five times larger than the ideal width. Results are compared against a typical tidal range in order to show uncertainty of predicted intertidal high width, intertidal low width and subtidal width as a fraction of the total estuary width.

## 4    Discussion

Results from this study illustrate that bed level distributions of channel and bar patterns in estuaries are topographically forced. The estuary outline that is observable from the surface translates into the three dimensional patterns below the water surface. Bar built estuaries typically have a quasi-periodic planform (Leuven et al., 2017), in which major channel confluences occur

at locations where the estuary is close to its ideal shape (Kleinhans et al., 2018; Leuven et al., 2018). The parts between the confluences are typically filled with intertidal bar complexes.

Previously, hypsometry was used to summarise the geometry of entire tidal basins or estuaries (Boon and Byrne, 1981; Dieckmann et al., 1987; Townend, 2008), but this may be an oversimplification for estuaries that are relatively long. These estuaries typically have a linear bed profile varying from an along-channel constant depth to strongly linear sloping (e.g.

the Mersey in UK). In the latter case, the elevation at which subtidal and intertidal area occur varies significantly along-channel (Blott et al., 2006). Additionally, friction and convergence may cause the tidal range to either dampen or amplify causing variation in tidal elevation, subtidal area and tidal prism (Savenije, 2006). Nevertheless, if desired, along-channel varying hypsometry predictions can be converted in one single summarising curve (Fig. 12), which shows that also the basin hypsometry can be predicted when limited data is available.

Our results show that hypsometry is not only a tool to predict morphology when limited data is available, but that hypsometry can also be used to reduce a large dataset of bathymetry and to study evolution of bathymetry over time. In the case of Strahler (1952), hypsometry fits result in along-channel profiles of $z$- and $r$-values, but in practice any function or shape could be fitted. For example, the locations along the Western Scheldt where major dredging and dumping took place showed weaker correlation





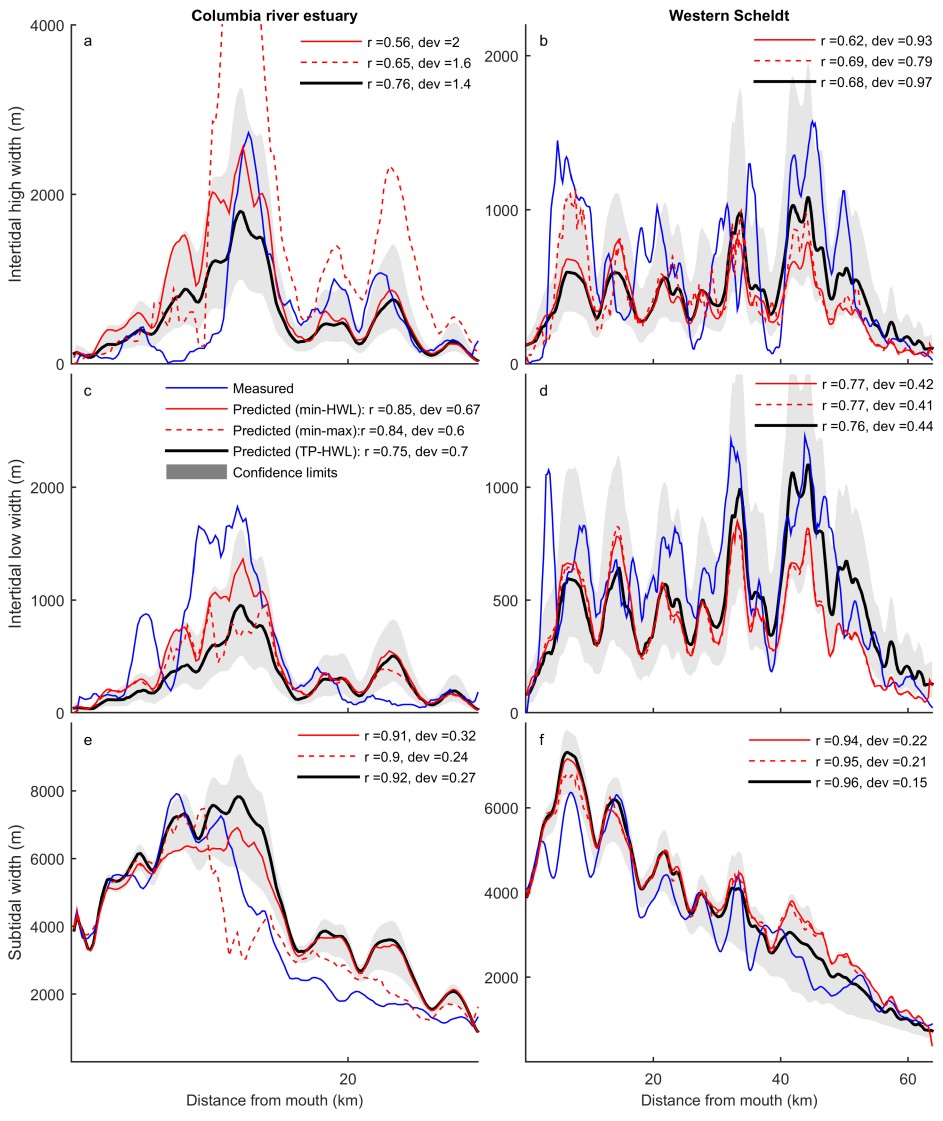

**Figure 10.** Comparison of measured and predicted values of intertidal high, intertidal low and subtidal width for the Columbia river estuary (a,c,e) and the Western Scheldt (b,d,f). Predicted nondimensional hypsometry was dimensionalised for each cross-section using three methods (explained in methods) and uncertainty margins are given for one of the predictions (solid black line). In the legend, $r$ indicates the Pearson product-moment correlation coefficient and $dev$ the average factor of deviation between the predicted (TP-HWL) and measured lines.





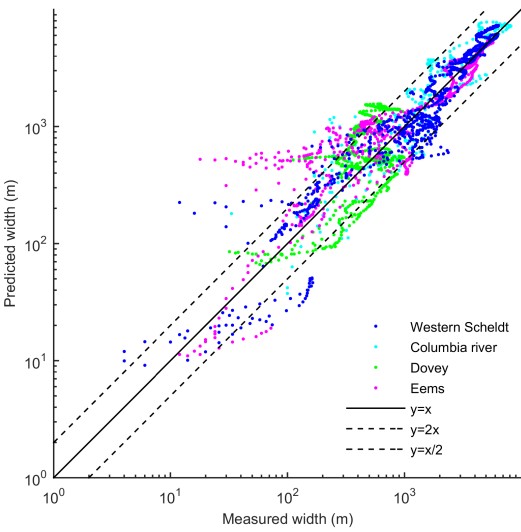

**Figure 11.** Comparison of measured and predicted width of intertidal and subtidal width. The (solid) line of equality indicates a perfect fit and dashed lines indicate a deviation of a factor 2. Percentage of measurements within these margins are indicated in Table 2.

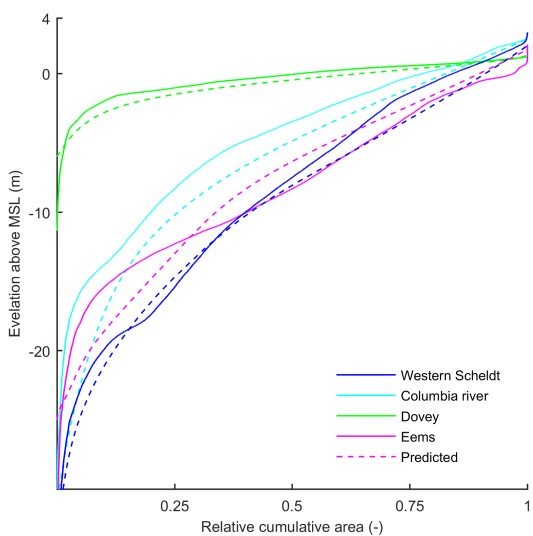

**Figure 12.** Hypsometry as summarised in a single curve for the entire estuaries. Solid lines are measured from bathymetry, dashed lines based on the predictions.





with the original Strahler (1952) shape (Fig. 13). In these cases, fits with higher quality (lower RMSE) were obtained when we used the inverted hypsometric function (Eq. 4) (Fig. 13b,e). So in practice, one could fit a range of different hypsometry shapes and subsequently find out which of these shapes fit best on the used dataset. It can indicate that certain parts along the estuary require a separate hypsometric description. The fitting parameters are a method to describe the along-channel

variation (Fig. 13b,d). Hypsometry can be fitted to compare data from nature, physical experiments and numerical modelling and subsequently study for example the effect of vegetation, cohesive mud and the influence of management on these systems.

### 4.1  Implications for management of estuaries

In many estuaries from around the world subtidal channels are used as shipping fairways, while the intertidal bars (or shoals) form valuable ecological habitat (e.g. Bouma et al., 2005). For example in the Western Scheldt, the shipping fairway is now

maintained at a depth of 14.5m below Lowest Astronomical Tide (de Vriend et al., 2011; Depreiter et al., 2012). With empirical hypsometry predictions we can estimate the width below a certain depth required for shipping, which gives estimates of what volume to dredge and at what locations along the estuary, which is relevant for construction of future shipping fairways in estuaries for which we may have limited data.

In contrast, low-dynamic intertidal areas are valuable ecological habitat, for example for the Western Scheldt there is an

obligation to maintain a certain amount of intertidal area (Depreiter et al., 2012). Previously, Townend (2008) showed that basin hypsometry can be a tool to design breaches in managed re-alignment sites and can provide an indication of habitat composition. Hypsometry analysis per cross-section shows that estuary outline translates into intertidal area, which implies that locations where the estuary is relatively wide have a relatively wide intertidal area. The ecological value is determined by the area of low-dynamical undeep water and intertidal areas (for settling and feeding) (Depreiter et al., 2012). This means

that the edges should neither become steeper nor higher (leading to permanent dry-fall) or deeper. Hypsometry fits (in case of available data) or predictions (in case of limited data) can indicate which locations along the estuary have a risk to transform away from low-dynamic area or have the potential to become low-dynamic area by suppletion of dredged sediment.

The occurrence of vegetation species depend on bed elevation, salinity, maximum flow velocity and sediment type (de Jong, 1999; Gurnell et al., 2012). Even though predicted hypsometry only gives bed elevations, a comparison of the height interval

in which *Salicornia* and *Spartina* can occur (Mckee and Patrick, 1988; Davy et al., 2001; van Braeckel et al., 2008), showed similar trends and the same order of magnitude as the measured vegetation from ecotope maps of the Western Scheldt in 2012 (Fig. 14). Some underpredictions arise in parts along the estuary where bed elevations above the high water level occur, such as at the Drowned land of Saeftinghe. However, in general the vegetation width is overpredicted because (1) hypsometry is stretched between the high water line and channel depth and (2) other constraining biotic and abiotic factors were excluded.

## 5  Conclusions


We studied the relation between along-channel planform geometry of sandy estuaries and their hypsometry, which characterises the distribution of along-channel and cross-channel bed-levels. The vertical dimensions were found to relate to the horizontal





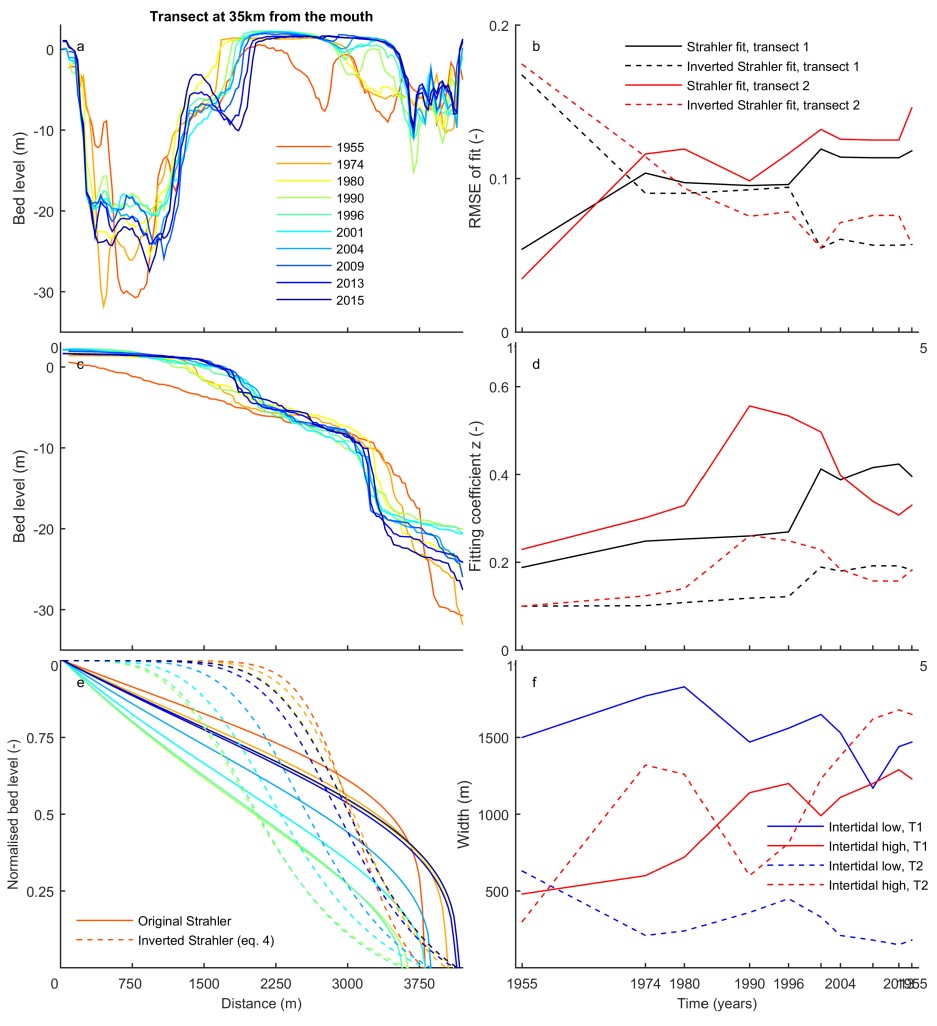

**Figure 13.** (a) Evolution of cross-section in the Western Scheldt where significantly has been dredged and dumped (The *Drempel van Hansweert*), (c) Measured hypsometric profiles of the same time steps. (e) Best fit hypsometries using original Strahler equation [solid] and inverted Strahler function (Eq. 4) [dashed], (b) Quality of the two types of fits shows that the shape of the best fitting hypsometric curve changes from convex to the inverted equation in the 1970s-1980s. (d) Fitting coefficients for z increase over time for both hypsometry types and both transects. (f) Intertidal high area increased over this period while intertidal low area remained constant [Transect 1] or decreased [Transect 2].




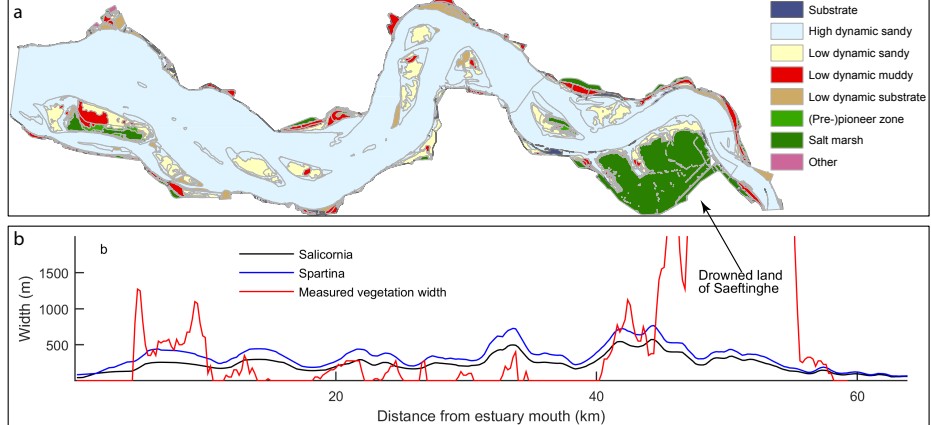

**Figure 14.** (a) Ecotope map of the Western Scheldt (2012), obtained from Rijkswaterstaat. (b) Prediction of the width in which *Salicornia* (black) and *Spartina* (blue) can occur when assuming that *Spartina* occurs between MSL and 1.5 m above and *Salicornia* occurs between 1.0 and 2.5m, while ignoring velocity, salinity and sediment type constraints. Red line indicates measured width of vegetation based on ecotope map. The Drowned land of Saeftingen is excluded in the predictions, because the high water line was the boundary of the analysed bathymetry, while it is included in the measured data.

dimensions. In other words, the degree to which the estuary width deviates from an ideal converging estuary shape reflects in the occurring hypsometry. At locations where the width is much larger than ideal, convex hypsometric shapes are observed, contrary to the locations where the estuary width is close to ideal, where concave hypsometric shapes are observed. In between these extreme end members, a gradual transition with quasi-periodic variation was observed. This implies that it is possible to predict the along-channel varying hypsometry of estuaries, which is relevant for estuaries for which limited data is available. To obtain broad brush estimates of the occurring bed levels, only the estuary outline and a typical tidal amplitude are required. The predictions can be used to study the presence and evolution of intertidal area, which forms valuable ecological habitat, and to get estimates of typical volumes that might need to be dredged when constructing shipping fairways.

*Data availability.* Bathymetry was obtained from Rijkswaterstaat for the Western Scheldt and Eems estuary, from Natural Resources Wales via Dr. Emmer Litt for the Dovey and from Lower Columbia Estuary Partnership for the Columbia River estuary.

*Author contributions.* The authors contributed in the following proportions to conception and design, data collection and processing, analysis and conclusions, and manuscript preparation: JRFWL(60,55,75,70%), SS(15,40,15,15%), MGK(25,5,10,15%).

*Competing interests.* The authors declare that they have no conflict of interest.

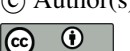



*Acknowledgements.* This research was supported by Future Deltas, Utrecht University (grant to JRFWL) and by the Dutch Technology Foundation TTW (grant Vici 016.140.316/13710 to MGK), which is part of the Netherlands Organisation for Scientific Research (NWO), and is partly funded by the Ministry of Economic Affairs. This work is part of the PhD research of JRFWL. We acknowledge a data processing contribution by Andy Bruijns as part of his MSc guided research. Reviewers will be acknowledged.




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
