# Peer review of "Morphology of bar-built estuaries: empirical relation between planform shape and depth distribution"

_Earth Surface Dynamics, 2018_

## Referee Comment (RC1) · I.H. Townend (Referee) · 13 Mar 2018

This paper provides a useful examination of along-channel variations in channel width hypsometry. The paper is well organised and clearly written. The data used and method of analysis are, in themselves, sound. However, I would like to suggest a few changes that would give the paper a more precise focus. These relate to the methodology and what it can be said to be examining.

The method of Strahler is adopted without any substantive explanation. However the Strahler equation was proposed for terrestrial landscapes and is based on plan areas as a function of elevation. The paper considers submerged (or at times partially submerged) bodies in terms of the cross-section width. The basis of this transposition is

not explained and the definitions of the terms in Equation 1 are not particularly clear. My reading is that 'h' is the proportion of total section height, and that 'y' is the proportion of the total section width. This does however omit the basis of r (which is a function of minimum and maximum plan area in Strahler) and makes it a fit parameter. This is useful strategy but Equation 1 is now simply a fitted shape function.

In the literature other authors (e.g. Boon and Byrne, 1981; and Townend, 2008) have adapted Strahler for use in the marine environment. The authors here have preferred the original (terrestrially based) Strahler equation. Given that they are all empirical relationships this may be entirely appropriate but some discussion as to why would provide a stronger link with the existing literature.

In the light of the above, I would suggest that it might also be appropriate to add the word empirical to both the title and the section entitled 'Relation between morphology and hypsometry'.

My other main concern relates to the use of the word 'ideal' in relation to the width of the channel. The study is essentially a geometric one, extracting width information from detailed bathymetries in four estuaries. Without consideration of some other metric such as tidal elevation/velocity, energy dissipation or the energy flux in the system it is not possible to assert a "state" of the system relative to equilibrium and hence to define what constitutes an "ideal" system, as classically defined. Whilst the authors make clear how they have defined their ideal plan form (width at the mouth and river) this only serves to compound a prevailing myth that the ideal is based on convergent width. If the cross-sectional area is exponentially convergent the estuary meets the basis of Pillsbury's original definition for an ideal estuary. If it happens that the hydraulic depth is constant along the channel then the CSA convergence length equates to the width convergence length.

There is some evidence from UK estuaries that width-depth variations provide a degree of system redundancy, allowing the system to adapt and so do minimum work, whilst

maintaining the CSA convergence. This is illustrated in the attached figure for the Humber, where the CSA is clearly exponentially convergent. The corresponding width and depth values vary about the exponential fits (seemingly in an inverse manner that it has been suggested is linked to overall channel sinuosity). Importantly in this context the width is invariably narrower and deeper at the mouth for a number of reasons (geology, drift, etc). Consequently, I would reason that the authors have examined the variance from the minimal width convergence. This does not detract from the results but it is important not to confuse a valid conclusion relating to along channel variation in width hypsometry, with assertions relating to an ideal system and its state relative to equilibrium. For the latter, I am of the opinion that we need a physically based determination of the hypsometry, rather than an empirical one.

Finally a point of detail. In the discussion, you refer to whole system hypsometry as an oversimplification. However, these whole system descriptions are consistent with the original Strahler concept of a basin hypsometry based on plan area. In a landform context these remain entirely valid descriptions. In terms of estuary dynamics they do not capture the along channel variations. As you note, there can be a significant variation of the high a low water surfaces along the estuary. Consequently, the along-channel cross-section hypsometry should not be assumed to be relative to a fixed vertical datum. Interpreting these along channel variations remains an open question because of the reasons outlined above.
* * *
[Figure]

[Figure]

[Figure]

**Fig. 1.** Illustration of along-channel width depth variation

---

## Referee Comment (RC2) · Anonymous Referee #2 · 5 Apr 2018

The paper investigates the relationship between estuary planform shape and along-channel variations in hypsometry. The authors recall the definition of "ideal estuary" and assume that along-channel changes in hypsometry (e.g. changes from concave to convex hypsometry) depend on the deviation of estuarine cross-sectional width from the "ideal width" dictated by an exponentially decreasing function. The paper builds upon previous findings by the authors (Leuven et al., 2016, 2017) showing that "excess width" (with respect to the ideal width) allows one to predict the location of tidal bars within the estuary. The new finding is that concave hypsometry occurs where no bars are observed and the estuary width is close to the ideal one, whereas convex hypsometry occurs where extensive bars develop at a given location (or cross section) and estuary width is much larger than the ideal one. The paper is well written and

clearly organized. It addresses a relevant issue of practical importance, particularly in view of current anthropogenic influence on estuarine morphology and dynamics. As such, it deserves credit and it will be of interest to the readership of ESurf. I have a few minor suggestions made in the effort to improve an already good paper. General Comments The authors use the "ideal estuary" model that is based on a set of assumptions. The authors then discuss their results by relating them to the ratio between the observed estuary width and the "ideal" width obtained by considering an exponential width variation along the estuary. As noted, the "ideal estuary" model embeds a set of assumptions that should be discussed more in detail. As an example, the authors compare "ideal" and observed widths, but then assume a linear landward decrease in channel depth, whereas the ideal model prescribes a different behavior. As to the use of hypsometry, it should be better clarified, from the very beginning, that the theoretical framework is quite different from the one proposed for river basins (Strahler, 1952) and for tidal basins (Boon and Byrne, 1981) because in this case the hypsometric curve is applied across channel width (it is a cross-sectional hypsometric curve). I find the idea clever and interesting, but I'd like to see some more discussion on the reasons leading the authors to set up such an analysis. In addition, in the case of the river and tidal basin, the morphological evolution was accounted for, suggesting that different shapes of the hypsometric curve were associated to young or old systems. Is there any possibility of making such an analogy within the framework proposed by the authors? Can the framework account for the dynamic nature of estuarine landscapes? I also wonder is the framework could be applied to any type of estuary of if there are some limitations. Can micro- and macrotidal systems behave in a different way? Finally, I remembered of a paper proposing quite a similar analysis (Toffolon and Crosato, JCR 2017). I think the paper would benefit from recalling the results of the above paper (analyses were applied to the Scheldt estuary). In that paper, the authors analyzed the case of U-shaped, V-shaped, Y-shaped cross section. This could be done also within this framework, to predict the tendency of the estuary to develop particular shapes. Detailed comments Page 1, Line 19 change "hydrodynamical" to "hydrodynamic" Page 1, Line 22. It should

be clarified that the loss of tidal energy by friction is balanced by the gain in tidal energy by convergence Page 4, Line 5. "width" should be "with" Page 5, Lines 4-5. Actually, this could be the other way round: the presence of bars generates excess width. Page 6 line 1 and line 5. I do not think there is the need to recall that "e" is Euler's number (actually, e^(*-x/Lw) is an exponential function) and "ln" is the natural logarithm. Page 6 line 8. Computation of channel width at the landward limit is unclear. Please explain. Page 7 lines 16-20. These lines should be rephrased. If I understood correctly, in the first case, fitting is performed on both r and z (as in the third case with the inverted function). Page 7 lines 25-27. Please discuss why the inverted function was used. Page 9 lines 5-7. The linear decrease in water depth from the mouth to the landward section is an assumption that needs be discussed (also in view of other theoretical frameworks developed for tidal channels, e.g. Toffolon and Lanzoni, JGR 2010). In addition, is such an assumption consistent with those embedded in the "ideal estuary" model? Page 9 equation (5). Please note that computing the tidal prism by multiplying estuary surface area by the tidal range tantamount to assume a flat water surface elevation along the estuary and moreover does not account for the fact that portions of the estuary area A(t) might get dry during the tidal cycle (see Boon, 1975). Figure 3. This figure should be modified. In my view it is a bit confusing to use the same axes for the two columns of panels. The left column should have plots with "bed elevation" on the vertical axis, while the right one should have h_z. Page 10, Figure 5. How was the typical profile for both cases obtained? Please clarify. Page 10, Caption of Figure 5. "disected" should be "dissected". Page 10, line 6. "suggest" should be "suggests". Page 12, line 6. "In general, the width at the mouth of the estuary and at the upstream estuary is close to ideal ..." shouldn't this be straightforward, due to the fact that you impose those BCs in eq. (2) to compute the ideal along-channel width? Please clarify. Page 12, lines 26-30. The reader might wonder why the predictor equation was not applied to the other two estuaries analysed in the manuscript. Page 15 line 5. I find it difficult to support and discuss the results by citing papers that are still in review or in preparation. Please remove, provide other references or update.

**ESurfD**

Interactive
comment

---

## Author Comment (AC1) · 30 Apr 2018

This paper provides a useful examination of along-channel variations in channel width hypsometry. The paper is well organised and clearly written. The data used and method of analysis are, in themselves, sound. However, I would like to suggest a few changes that would give the paper a more precise focus. These relate to the methodology and what it can be said to be examining.

*We found the review helpful and positive and thank the reviewer in the acknowledgements. Below we describe (in italics) how we used the reviewer comments to improve the manuscript.*

The method of Strahler is adopted without any substantive explanation. However the Strahler equation was proposed for terrestrial landscapes and is based on plan areas as a function of elevation. The paper considers submerged (or at times partially submerged) bodies in terms of the cross-section width. The basis of this transposition is not explained and the definitions of the terms in Equation 1 are not particularly clear. My reading is that 'h' is the proportion of total section height, and that 'y' is the proportion of the total section width. This does however omit the basis of r (which is a function of minimum and maximum plan area in Strahler) and makes it a fit parameter. This is useful strategy but Equation 1 is now simply a fitted shape function. In the literature other authors (e.g. Boon and Byrne, 1981; and Townend, 2008) have adapted Strahler for use in the marine environment. The authors here have preferred the original (terrestrially based) Strahler equation. Given that they are all empirical relationships this may be entirely appropriate but some discussion as to why would provide a stronger link with the existing literature.

*We now clarify in the text (1) why we adopted the Strahler formulation, (2) why the environment for which it was proposed is less relevant and (3) that we indeed made r and z fitting parameters and use Equation 1 as a fitting function. The text now reads: "In the past, multiple authors have proposed empirical relations for the hypsometric shape of terrestrial landscapes (Strahler 1952) and (partially) submerged bodies (Boon, 1981; Wang, 2002; Toffolon, 2007; Townend, 2008) (see Townend, 2008 for review). All equations, except for Wang (2002), predict a fairly similar hypsometric curve based on the volume and height range of the landform (Townend, 2008). While it is of interest to use these empirical relations to predict the occurring altitude variation of a landform, the framework here is different, because in this case the hypsometric curve is applied across channel width: it is a cross-sectional hypsometric curve. We aim to use the general hypsometric curve to characterise the occurring cross-sectional hypsometry, which is similar to the approach of Toffolon & Crosato (2007) who fitted a power function to 15 zones along the Western Scheldt. To that end, it is less relevant for which environment the hypsometric relation was proposed, as long as it is capable to describe the range of occurring hypsometries. For the case of the estuarine environment (Fig. 3), the hypsometric curve should be able to describe variations in concavity and variations in the slope of the curve at the inflection point. Here we use the original (Strahler 1952) formulation, which is capable to do so, but in principle any equation that fits well could be used."*

*After the Strahler equation we added the reviewers suggestion that 'h' is the proportion of total section height, that 'y' is the proportion of the total section width and that our approach changes the definition of r (which is a function of minimum and maximum plan area in Strahler) to make it a fitting parameter.*

In the light of the above, I would suggest that it might also be appropriate to add the word empirical to both the title and the section entitled 'Relation between morphology and hypsometry'.
*We added the word 'empirical' in the manuscript title and section title, as well as in the abstract text.*

My other main concern relates to the use of the word 'ideal' in relation to the width of the channel. The study is essentially a geometric one, extracting width information from detailed bathymetries in four estuaries. Without consideration of some other metric such as tidal elevation/velocity, energy dissipation or the energy flux in the system it is not possible to assert a "state" of the system relative

to equilibrium and hence to define what constitutes an "ideal" system, as classically defined. Whilst the authors make clear how they have defined their ideal plan form (width at the mouth and river) this only serves to compound a prevailing myth that the ideal is based on convergent width. If the cross-sectional area is exponentially convergent the estuary meets the basis of Pillsbury's original definition for an ideal estuary. If it happens that the hydraulic depth is constant along the channel then the CSA convergence length equates to the width convergence length.

*The figure below shows the along-channel profiles of width, width-averaged depth and cross-sectional area for the systems we studied. Cross-sectional area profiles are rather linear than exponentially convergent and along-channel depth profiles are rarely constant, so the estuaries deviate from Pillsbury's original definition for an ideal estuary, but it is precisely the effect of deviation from an ideal shape on bar patterns and bed levels that we are interested in.*

*The reviewer comments that the equilibrium ideal state of an estuary might be confused with the geometric ideal width that we use in our study. To prevent any confusion between the equilibrium estuary state and the geometric width profile, we now clarify the definition of an ideal estuary in a separate section at the start of the methods and explicitly state its relation with geometric properties. Given that channel width is the only geometric property that we can measure from aerial photography we subsequently explain that deviation from an ideal converging width profile is the only way we can approximate deviation from an ideal estuary shape.*

*The new text reads:*
*"A useful model to describe the morphology of estuaries is that of the 'ideal estuary'. In an ideal estuary the energy per unit width remains constant along estuarine channels. This ideal state can be met when tidal range and tidal current are constant along-channel, such that the loss of tidal energy by friction is balanced by the gain in tidal energy per unit width by channel convergence (Pillsbury, 1956; Dronkers, 2017). In case the depth is constant along the channel, the ideal estuary conditions are approximately met when the width is exponentially decreasing in landward direction (Pillsbury, 1956; Langbein, 1963; Savenije, 2006; Toffolon & Crosato, 2010; Savenije, 2015), which also implies an along-channel converging cross-sectional area. However, when depth and friction are not constant along-channel, but for example linearly decreasing in landward direction, less convergence in width is required to maintain constant energy per unit of width. Many natural estuaries are neither in equilibrium nor in a condition of constant tidal energy per unit width. They deviate from the ideal ones as result of varying degree of sediment supply, lack of time for adaptation to changing upstream conditions and sea-level rise (Townend, 2012; de Haas et al., 2017). Whether continued sedimentation would reform bar-built estuaries into proper ideal estuaries remains an open question. For our application, the concept of ideal estuaries is useful to assess the degree of deviation from ideal because of the width variations observed as bars, tidal flats and saltmarsh. While we expect a somewhat different degree of convergence such that the ideal state of constant energy per unit of width is approximately maintained, we do not study the deviation of this convergence length from that in ideal estuaries.*

*Ideally, we would want to assess the degree to which an estuary is in equilibrium from an aerial photograph, because this is often the only data available. However, the only indicator derivable from aerial photography is channel width and thus deviation from a converging width profile. Therefore, in Leuven et al. (2017), we defined the excess width, which is the local width of the estuary minus our approximation of the potential ideal estuary width. Here, the ideal estuary width is approximated as an exponential fit on the width of the mouth and the width of the landward river. While the empirical measure of 'ideal width' should not be confused with the 'ideal state' of an estuary, it is the only practical way to estimate deviation from an ideal estuary based on the estuary outline only. Moreover, it proved to be a good indicator of occurring bar patterns (Leuven et al., 2017) and will therefore be applied in this paper to study hypsometries."*

[Figure]

*We considered alternative wording for ideal width in the remainder of the manuscript. Because we now explain how we derived this geometrical property from the concept of an ideal shape and explicitly state that it should not be confused with the equilibrium state, we decided to keep te wording of ideal width. Moreover (1) it is precisely the deviation from an ideal shape that we are interested in, (2) the use of this terminology is in agreement with previous work (Leuven et al., 2017) and (3) other wordings that we considered might lead to misunderstanding, e.g. minimal width convergence can be read as minimal convergence. If the editor prefers different, we will consider the use of minimal width convergence, convergence of minimum width or something similar.*

There is some evidence from UK estuaries that width-depth variations provide a degree of system redundancy, allowing the system to adapt and so do minimum work, whilst maintaining the CSA convergence. This is illustrated in the attached figure for the Humber, where the CSA is clearly exponentially convergent. The corresponding width and depth values vary about the exponential fits (seemingly in an inverse manner that it has been suggested is linked to overall channel sinuosity). Importantly in this context the width is invariably narrower and deeper at the mouth for a number of reasons (geology, drift, etc). Consequently, I would reason that the authors have examined the variance from the minimal width convergence. This does not detract from the results but it is important not to confuse a valid conclusion relating to along channel variation in width hypsometry, with assertions relating to an ideal system and its state relative to equilibrium. For the latter, I am of the opinion that we need a physically based determination of the hypsometry, rather than an empirical one.

*Thank you, we agree and this case agrees with our findings. See reply to comment above about confusion of ideal width with ideal system state. We now clarify this in a separate section in the methods.*

*As for the suggestion that we need a physically based determination of the hypsometry: this is the ultimate aim that is presently beyond reach. We added a paragraph to the discussion about this idea, which reads: "Here we found that the cross-sectional hypsometry relates to occurring bar patterns and estuarine geometry. In contrast to an empirical description, ideally, a physics-based determination of the hypsometry would be favourable. However, with the current state of the art of bar theory (Leuven et al., 2016) and relations for intertidal area, tidal prism, cross-sectional area and flow velocities (O'Brien, 1969; Friedrichs & Aubrey, 1988) it is not yet possible to derive a theoretical prediction of hypsometry. For example, bar theory (Seminara & Tubino, 2001; Schramkowski et al., 2002) could predict occurring bar patterns on top of an (ideal) estuary shape, but current theories overpredict their dimensions (Leuven et al., 2016) and it is still impossible to scale these to bed level variations, because the theories are linear. In addition to that, the resulting predictions would need to meet the requirement that the predicted bed levels and the intertidal area together lead to hydrodynamic conditions that fit the estuary as well."*

Finally a point of detail. In the discussion, you refer to whole system hypsometry as an oversimplification. However, these whole system descriptions are consistent with the original Strahler concept of a basin hypsometry based on plan area. In a landform context these remain entirely valid descriptions. In terms of estuary dynamics they do not capture the along channel variations. As you note, there can be a significant variation of the high a low water surfaces along the estuary. Consequently, the along-channel cross-section hypsometry should not be assumed to be relative to a fixed vertical datum. Interpreting these along channel variations remains an open question because of the reasons outlined above.

*We added the suggestions of the reviewer to the paragraph about the degree to which whole system hypsometry are oversimplifications for estuaries. The paragraph now reads, with bold parts added:*

*"Previously, hypsometry was used to summarise the geometry of entire tidal basins or estuaries (Boon 1981; Dieckmann 1987; Townend 2008).* **The whole system descriptions are consistent with the original Strahler (1952) concept of a basin hypsometry based on plan area, which is a valid description in a landform context.** *However, these descriptions oversimplify* **the along-channel variability** *in estuaries that are relatively long. These estuaries typically have a linear bed profile varying from an along-channel constant depth to strongly linear sloping (e.g. the Mersey in UK). In the latter case, the elevation at which subtidal and intertidal area occur varies significantly along-channel (Blott 2006). Additionally, friction and convergence may cause the tidal range to either dampen or amplify causing variation in tidal elevation, subtidal area and tidal prism (Savenije 2006).* **Consequently, the along-channel cross-section hypsometry should be assumed to be relative to an along-channel varying high water level or mean sea level rather than an along-channel fixed vertical datum. Interpreting these along channel variations remains an open question because of the reasons outlined above.** *Nevertheless, if desired, along-channel varying hypsometry predictions can be converted in one single summarising curve (Fig. 12), which shows that also the basin hypsometry can be predicted when limited data is available."*

---

## Author Comment (AC2) · 30 Apr 2018

RC2: 'Review', Anonymous Referee #2, 05 Apr 2018
The paper investigates the relationship between estuary planform shape and along-channel variations in hypsometry. The authors recall the definition of "ideal estuary" and assume that along-channel changes in hypsometry (e.g. changes from concave to convex hypsometry) depend on the deviation of estuarine cross-sectional width from the "ideal width" dictated by an exponentially decreasing function. The paper builds upon previous findings by the authors (Leuven et al., 2016, 2017) showing that "excess width" (with respect to the ideal width) allows one to predict the location of tidal bars within the estuary. The new finding is that concave hypsometry occurs where no bars are observed and the estuary width is close to the ideal one, whereas convex hypsometry occurs where extensive bars develop at a given location (or cross section) and estuary width is much larger than the ideal one. The paper is well written and clearly organized. It addresses a relevant issue of practical importance, particularly in view of current anthropogenic influence on estuarine morphology and dynamics. As such, it deserves credit and it will be of interest to the readership of ESurf. I have a few minor suggestions made in the effort to improve an already good paper.
*We found the review helpful and positive and thank the reviewer in the acknowledgements. Below we describe (in italics) how we used the reviewer comments to improve the manuscript.*

**General Comments**
The authors use the "ideal estuary" model that is based on a set of assumptions. The authors then discuss their results by relating them to the ratio between the observed estuary width and the "ideal" width obtained by considering an exponential width variation along the estuary. As noted, the "ideal estuary" model embeds a set of assumptions that should be discussed more in detail. As an example, the authors compare "ideal" and observed widths, but then assume a linear landward decrease in channel depth, whereas the ideal model prescribes a different behavior.
*This comment relates to the third comment of reviewer 1. We now added a new section at the start of the methods section in which we discuss the assumptions of the ideal estuary in more detail, including explanation how this lead to our geometric approach and that it shouldn't be confused with the "ideal" state:*

*"A useful model to describe the morphology of estuaries is that of the 'ideal estuary'. In an ideal estuary the energy per unit width remains constant along estuarine channels. This ideal state can be met when tidal range and tidal current are constant along-channel, such that the loss of tidal energy by friction is balanced by the gain in tidal energy per unit width by channel convergence (Pillsbury, 1956; Dronkers, 2017). In case the depth is constant along the channel, the ideal estuary conditions are approximately met when the width is exponentially decreasing in landward direction (Pillsbury, 1956; Langbein, 1963; Savenije, 2006; Toffolon & Crosato, 2010; Savenije, 2015), which also implies an along-channel converging cross-sectional area. However, when depth and friction are not constant along-channel, but for example linearly decreasing in landward direction, less convergence in width is required to maintain constant energy per unit of width. Many natural estuaries are neither in equilibrium nor in a condition of constant tidal energy per unit width. They deviate from the ideal ones as result of varying degree of sediment supply, lack of time for adaptation to changing upstream conditions and sea-level rise (Townend, 2012; de Haas et al., 2017). Whether continued sedimentation would reform bar-built estuaries into proper ideal estuaries remains an open question. For our application, the concept of ideal estuaries is useful to assess the degree of deviation from ideal because of the width variations observed as bars, tidal flats and saltmarsh. While we expect a somewhat different degree of convergence such that the ideal state of constant energy per unit of width is approximately maintained, we do not study the deviation of this convergence length from that in ideal estuaries.*

*Ideally, we would want to assess the degree to which an estuary is in equilibrium from an aerial photograph, because this is often the only data available. However, the only indicator derivable from aerial photography is channel width and thus deviation from a converging width profile. Therefore, in*

*Leuven et al. (2017), we defined the excess width, which is the local width of the estuary minus our approximation of the potential ideal estuary width. Here, the ideal estuary width is approximated as an exponential fit on the width of the mouth and the width of the landward river. While the empirical measure of 'ideal width' should not be confused with the 'ideal state' of an estuary, it is the only practical way to estimate deviation from an ideal estuary based on the estuary outline only. Moreover, it proved to be a good indicator of occurring bar patterns (Leuven et al., 2017) and will therefore be applied in this paper to study hypsometries."*

*It should be noted here as well that a linear along-channel depth profile can also be a horizontal bed profile in the case that the predicted channel depth based on hydraulic geometry at the landward side and the predicted depth at the mouth based on tidal prism-CSA relations is equal. We clarified this in the text: "Width-averaged depth profiles along estuaries are often (near-) linear (Savenije, 2015; Leuven et al., 2017), which includes horizontal profiles with constant depth."*

As to the use of hypsometry, it should be better clarified, from the very beginning, that the theoretical framework is quite different from the one proposed for river basins (Strahler, 1952) and for tidal basins (Boon and Byrne, 1981) because in this case the hypsometric curve is applied across channel width (it is a cross-sectional hypsometric curve). I find the idea clever and interesting, but I'd like to see some more discussion on the reasons leading the authors to set up such an analysis. In addition, in the case of the river and tidal basin, the morphological evolution was accounted for, suggesting that different shapes of the hypsometric curve were associated to young or old systems. Is there any possibility of making such an analogy within the framework proposed by the authors? Can the framework account for the dynamic nature of estuarine landscapes? I also wonder if the framework could be applied to any type of estuary of if there are some limitations. Can micro- and macrotidal systems behave in a different way?

*We now clarified that our approach is different from classical hypsometry studies in the first paragraph of the section about the general hypsometric curve: "(…) While it is of interest to use these empirical relations to predict the occurring altitude variation of a landform, the framework here is different, because in this case the hypsometric curve is applied across channel width: it is a cross-sectional hypsometric curve that we change along the system. We aim to use the general hypsometric curve to characterise the occurring cross-sectional hypsometry, (…)".*

*We added a paragraph in the discussion about need for a physically based determination of the hypsometry, and the lack of theory available to do so, also suggested by reviewer 1. A more physics based theory for hypsometry would be required to answer the open questions proposed by the reviewer.*

Finally, I remembered of a paper proposing quite a similar analysis (Toffolon and Crosato, JCR 2017). I think the paper would benefit from recalling the results of the above paper (analyses were applied to the Scheldt estuary). In that paper, the authors analyzed the case of U-shaped, V-shaped, Y-shaped cross section. This could be done also within this framework, to predict the tendency of the estuary to develop particular shapes.

*We were aware of this paper but showed in our earlier work that their application of bar theory is flawed. Indeed, they used a similar fitting approach as we did, but (1) used a power function instead of the Strahler formulation and (2) fitted hypsometric profiles for 15 zones along the estuary and characterise their shape. The effect is that they completely smooth out all differences between bars and channel zones which is precisely the point of our paper. Their U-shaped profiles correspond to our concave profiles, Y-shaped to our convex and V-shaped is exactly intermediate. We added references to their methodology in the introduction and methods section of our paper and compare the results of both studies in the discussion.*

Introduction: *"Furthermore, hypsometry was used as a data reduction method **to characterise entire reaches spanning bars and channels in estuaries (Toffolon & Crosato, 2007} and** shapes of individual tidal bar tops (de Vet et al., 2017)."*

*Methods:* "While it is of interest to use these empirical relations to predict the occurring altitude variation of a landform, the framework here is different, because in this case the hypsometric curve is applied across channel width: it is a cross-sectional hypsometric curve that we change along the system. We aim to use the general hypsometric curve to characterise the occurring cross-sectional hypsometry, which is similar to the approach of Toffolon & Crosato (2007) who fitted a power function to 15 zones along the Western Scheldt. However, the zoned approach smooths out all differences between bar complex and channel-dominated zones, which are of interest for this study."

*Discussion:* "These findings are consistent with hypsometry zonations previously found for the Western Scheldt with more concave hypsometries in for channel-dominated morphology and more convex hypsometries for bar complex morphology (Toffolon & Crosato, 2007). Our cross-sectional approach additionally revealed quasi-periodic behaviour within these zones."

**Detailed comments**

*Detailed textual comments were mostly incorporated and sometimes used as indicator where text clarity had to be improved. We attach a PDF that highlights the changes to the first submission.*

---

## Author Comment (AC3) · 30 Apr 2018

To the Editor and Associate Editor,

Please find enclosed the revised manuscript 'Morphology of bar-built estuaries: empirical relation between planform shape and depth distribution' by authors Jasper R.F.W. Leuven, Sanja Selaković and Maarten G. Kleinhans.

We found the reviews helpful and positive. We thank the reviewers in the acknowledgements. All reviewers' suggestions are now implemented in the new version of the manuscript.

Below we describe (in *italics*) how we used the reviewer comments to improve the manuscript. Detailed textual comments were mostly incorporated and sometimes used as indicator where text clarity had to be improved. We attach a PDF that highlights the changes to the first submission and refer to the changes below.

Kind regards,
Jasper Leuven (on behalf of all authors)

**RC1: 'Reviewers comments for esurf-2018-18', Ian Townend, 13 Mar 2018**

This paper provides a useful examination of along-channel variations in channel width hypsometry. The paper is well organised and clearly written. The data used and method of analysis are, in themselves, sound. However, I would like to suggest a few changes that would give the paper a more precise focus. These relate to the methodology and what it can be said to be examining.
*We found the review helpful and positive and thank the reviewer in the acknowledgements. Below we describe (in italics) how we used the reviewer comments to improve the manuscript.*

The method of Strahler is adopted without any substantive explanation. However the Strahler equation was proposed for terrestrial landscapes and is based on plan areas as a function of elevation. The paper considers submerged (or at times partially submerged) bodies in terms of the cross-section width. The basis of this transposition is not explained and the definitions of the terms in Equation 1 are not particularly clear. My reading is that 'h' is the proportion of total section height, and that 'γ' is the proportion of the total section width. This does however omit the basis of r (which is a function of minimum and maximum plan area in Strahler) and makes it a fit parameter. This is useful strategy but Equation 1 is now simply a fitted shape function. In the literature other authors (e.g. Boon and Byrne, 1981; and Townend, 2008) have adapted Strahler for use in the marine environment. The authors here have preferred the original (terrestrially based) Strahler equation. Given that they are all empirical relationships this may be entirely appropriate but some discussion as to why would provide a stronger link with the existing literature.
*We now clarify in the text (1) why we adopted the Strahler formulation, (2) why the environment for which it was proposed is less relevant and (3) that we indeed made r and z fitting parameters and use Equation 1 as a fitting function. The text now reads: "In the past, multiple authors have proposed empirical relations for the hypsometric shape of terrestrial landscapes (Strahler 1952) and (partially) submerged bodies (Boon, 1981; Wang, 2002; Toffolon, 2007; Townend, 2008) (see Townend, 2008 for review). All equations, except for Wang (2002), predict a fairly similar hypsometric curve based on the volume and height range of the landform (Townend, 2008). While it is of interest to use these empirical relations to predict the occurring altitude variation of a landform, the framework here is different, because in this case the hypsometric curve is applied across channel width: it is a cross-sectional hypsometric curve. We aim to use the general hypsometric curve to characterise the occurring cross-sectional hypsometry, which is similar to the approach of Toffolon & Crosato (2007) who fitted a power function to 15 zones along the Western Scheldt. To that end, it is less relevant for which environment the hypsometric relation was proposed, as long as it is capable to describe the*

*range of occurring hypsometries. For the case of the estuarine environment (Fig. 3), the hypsometric curve should be able to describe variations in concavity and variations in the slope of the curve at the inflection point. Here we use the original (Strahler 1952) formulation, which is capable to do so, but in principle any equation that fits well could be used."*

*After the Strahler equation we added the reviewers suggestion that 'h' is the proportion of total section height, that 'y' is the proportion of the total section width and that our approach changes the definition of r (which is a function of minimum and maximum plan area in Strahler) to make it a fitting parameter.*

In the light of the above, I would suggest that it might also be appropriate to add the word empirical to both the title and the section entitled 'Relation between morphology and hypsometry'.
*We added the word 'empirical' in the manuscript title and section title, as well as in the abstract text.*

My other main concern relates to the use of the word 'ideal' in relation to the width of the channel. The study is essentially a geometric one, extracting width information from detailed bathymetries in four estuaries. Without consideration of some other metric such as tidal elevation/velocity, energy dissipation or the energy flux in the system it is not possible to assert a "state" of the system relative to equilibrium and hence to define what constitutes an "ideal" system, as classically defined. Whilst the authors make clear how they have defined their ideal plan form (width at the mouth and river) this only serves to compound a prevailing myth that the ideal is based on convergent width. If the cross-sectional area is exponentially convergent the estuary meets the basis of Pillsbury's original definition for an ideal estuary. If it happens that the hydraulic depth is constant along the channel then the CSA convergence length equates to the width convergence length.
*The figure below shows the along-channel profiles of width, width-averaged depth and cross-sectional area for the systems we studied. Cross-sectional area profiles are rather linear than exponentially convergent and along-channel depth profiles are rarely constant, so the estuaries deviate from Pillsbury's original definition for an ideal estuary, but it is precisely the effect of deviation from an ideal shape on bar patterns and bed levels that we are interested in.*

*The reviewer comments that the equilibrium ideal state of an estuary might be confused with the geometric ideal width that we use in our study. To prevent any confusion between the equilibrium estuary state and the geometric width profile, we now clarify the definition of an ideal estuary in a separate section at the start of the methods and explicitly state its relation with geometric properties. Given that channel width is the only geometric property that we can measure from aerial photography we subsequently explain that deviation from an ideal converging width profile is the only way we can approximate deviation from an ideal estuary shape.*

*The new text reads:*
*"A useful model to describe the morphology of estuaries is that of the 'ideal estuary'. In an ideal estuary the energy per unit width remains constant along estuarine channels. This ideal state can be met when tidal range and tidal current are constant along-channel, such that the loss of tidal energy by friction is balanced by the gain in tidal energy per unit width by channel convergence (Pillsbury, 1956; Dronkers, 2017). In case the depth is constant along the channel, the ideal estuary conditions are approximately met when the width is exponentially decreasing in landward direction (Pillsbury, 1956; Langbein, 1963; Savenije, 2006; Toffolon & Crosato, 2010; Savenije, 2015), which also implies an along-channel converging cross-sectional area. However, when depth and friction are not constant along-channel, but for example linearly decreasing in landward direction, less convergence in width is required to maintain constant energy per unit of width. Many natural estuaries are neither in equilibrium nor in a condition of constant tidal energy per unit width. They deviate from the ideal ones as result of varying degree of sediment supply, lack of time for adaptation to changing upstream conditions and sea-level rise (Townend, 2012; de Haas et al., 2017). Whether continued*

*sedimentation would reform bar-built estuaries into proper ideal estuaries remains an open question. For our application, the concept of ideal estuaries is useful to assess the degree of deviation from ideal because of the width variations observed as bars, tidal flats and saltmarsh. While we expect a somewhat different degree of convergence such that the ideal state of constant energy per unit of width is approximately maintained, we do not study the deviation of this convergence length from that in ideal estuaries.*

*Ideally, we would want to assess the degree to which an estuary is in equilibrium from an aerial photograph, because this is often the only data available. However, the only indicator derivable from aerial photography is channel width and thus deviation from a converging width profile. Therefore, in Leuven et al. (2017), we defined the excess width, which is the local width of the estuary minus our approximation of the potential ideal estuary width. Here, the ideal estuary width is approximated as an exponential fit on the width of the mouth and the width of the landward river. While the empirical measure of 'ideal width' should not be confused with the 'ideal state' of an estuary, it is the only practical way to estimate deviation from an ideal estuary based on the estuary outline only. Moreover, it proved to be a good indicator of occurring bar patterns (Leuven et al., 2017) and will therefore be applied in this paper to study hypsometries."*

[Figure]

*We considered alternative wording for ideal width in the remainder of the manuscript. Because we now explain how we derived this geometrical property from the concept of an ideal shape and explicitly state that it should not be confused with the equilibrium state, we decided to keep te*

*wording of ideal width. Moreover (1) it is precisely the deviation from an ideal shape that we are interested in, (2) the use of this terminology is in agreement with previous work (Leuven et al., 2017) and (3) other wordings that we considered might lead to misunderstanding, e.g. minimal width convergence can be read as minimal convergence. If the editor prefers different, we will consider the use of minimal width convergence, convergence of minimum width or something similar.*

There is some evidence from UK estuaries that width-depth variations provide a degree of system redundancy, allowing the system to adapt and so do minimum work, whilst maintaining the CSA convergence. This is illustrated in the attached figure for the Humber, where the CSA is clearly exponentially convergent. The corresponding width and depth values vary about the exponential fits (seemingly in an inverse manner that it has been suggested is linked to overall channel sinuosity). Importantly in this context the width is invariably narrower and deeper at the mouth for a number of reasons (geology, drift, etc). Consequently, I would reason that the authors have examined the variance from the minimal width convergence. This does not detract from the results but it is important not to confuse a valid conclusion relating to along channel variation in width hypsometry, with assertions relating to an ideal system and its state relative to equilibrium. For the latter, I am of the opinion that we need a physically based determination of the hypsometry, rather than an empirical one.

*Thank you, we agree and this case agrees with our findings. See reply to comment above about confusion of ideal width with ideal system state. We now clarify this in a separate section in the methods.*

*As for the suggestion that we need a physically based determination of the hypsometry: this is the ultimate aim that is presently beyond reach. We added a paragraph to the discussion about this idea, which reads: "Here we found that the cross-sectional hypsometry relates to occurring bar patterns and estuarine geometry. In contrast to an empirical description, ideally, a physics-based determination of the hypsometry would be favourable. However, with the current state of the art of bar theory (Leuven et al., 2016) and relations for intertidal area, tidal prism, cross-sectional area and flow velocities (O'Brien, 1969; Friedrichs & Aubrey, 1988) it is not yet possible to derive a theoretical prediction of hypsometry. For example, bar theory (Seminara & Tubino, 2001; Schramkowski et al., 2002) could predict occurring bar patterns on top of an (ideal) estuary shape, but current theories overpredict their dimensions (Leuven et al., 2016) and it is still impossible to scale these to bed level variations, because the theories are linear. In addition to that, the resulting predictions would need to meet the requirement that the predicted bed levels and the intertidal area together lead to hydrodynamic conditions that fit the estuary as well."*

Finally a point of detail. In the discussion, you refer to whole system hypsometry as an oversimplification. However, these whole system descriptions are consistent with the original Strahler concept of a basin hypsometry based on plan area. In a landform context these remain entirely valid descriptions. In terms of estuary dynamics they do not capture the along channel variations. As you note, there can be a significant variation of the high a low water surfaces along the estuary. Consequently, the along-channel cross-section hypsometry should not be assumed to be relative to a fixed vertical datum. Interpreting these along channel variations remains an open question because of the reasons outlined above.

*We added the suggestions of the reviewer to the paragraph about the degree to which whole system hypsometry are oversimplifications for estuaries. The paragraph now reads, with bold parts added:*
*"Previously, hypsometry was used to summarise the geometry of entire tidal basins or estuaries (Boon 1981; Dieckmann 1987; Townend 2008). **The whole system descriptions are consistent with the original Strahler (1952) concept of a basin hypsometry based on plan area, which is a valid description in a landform context.** However, these descriptions oversimplify **the along-channel variability** in estuaries that are relatively long. These estuaries typically have a linear bed profile varying from an along-channel constant depth to strongly linear sloping (e.g. the Mersey in UK). In*

*the latter case, the elevation at which subtidal and intertidal area occur varies significantly along-channel (Blott 2006). Additionally, friction and convergence may cause the tidal range to either dampen or amplify causing variation in tidal elevation, subtidal area and tidal prism (Savenije 2006).* **Consequently, the along-channel cross-section hypsometry should be assumed to be relative to an along-channel varying high water level or mean sea level rather than an along-channel fixed vertical datum. Interpreting these along channel variations remains an open question because of the reasons outlined above.** *Nevertheless, if desired, along-channel varying hypsometry predictions can be converted in one single summarising curve (Fig. 12), which shows that also the basin hypsometry can be predicted when limited data is available."*

**RC2: 'Review', Anonymous Referee #2, 05 Apr 2018**

The paper investigates the relationship between estuary planform shape and along-channel variations in hypsometry. The authors recall the definition of "ideal estuary" and assume that along-channel changes in hypsometry (e.g. changes from concave to convex hypsometry) depend on the deviation of estuarine cross-sectional width from the "ideal width" dictated by an exponentially decreasing function. The paper builds upon previous findings by the authors (Leuven et al., 2016, 2017) showing that "excess width" (with respect to the ideal width) allows one to predict the location of tidal bars within the estuary. The new finding is that concave hypsometry occurs where no bars are observed and the estuary width is close to the ideal one, whereas convex hypsometry occurs where extensive bars develop at a given location (or cross section) and estuary width is much larger than the ideal one. The paper is well written and clearly organized. It addresses a relevant issue of practical importance, particularly in view of current anthropogenic influence on estuarine morphology and dynamics. As such, it deserves credit and it will be of interest to the readership of ESurf. I have a few minor suggestions made in the effort to improve an already good paper.

*We found the review helpful and positive and thank the reviewer in the acknowledgements. Below we describe (in italics) how we used the reviewer comments to improve the manuscript.*

**General Comments**

The authors use the "ideal estuary" model that is based on a set of assumptions. The authors then discuss their results by relating them to the ratio between the observed estuary width and the "ideal" width obtained by considering an exponential width variation along the estuary. As noted, the "ideal estuary" model embeds a set of assumptions that should be discussed more in detail. As an example, the authors compare "ideal" and observed widths, but then assume a linear landward decrease in channel depth, whereas the ideal model prescribes a different behavior.

*This comment relates to the third comment of reviewer 1. We now added a new section at the start of the methods section in which we discuss the assumptions of the ideal estuary in more detail, including explanation how this lead to our geometric approach and that it shouldn't be confused with the "ideal" state:*

*"A useful model to describe the morphology of estuaries is that of the 'ideal estuary'. In an ideal estuary the energy per unit width remains constant along estuarine channels. This ideal state can be met when tidal range and tidal current are constant along-channel, such that the loss of tidal energy by friction is balanced by the gain in tidal energy per unit width by channel convergence (Pillsbury, 1956; Dronkers, 2017). In case the depth is constant along the channel, the ideal estuary conditions are approximately met when the width is exponentially decreasing in landward direction (Pillsbury, 1956; Langbein, 1963; Savenije, 2006; Toffolon & Crosato, 2010; Savenije, 2015), which also implies an along-channel converging cross-sectional area. However, when depth and friction are not constant along-channel, but for example linearly decreasing in landward direction, less convergence in width is required to maintain constant energy per unit of width. Many natural estuaries are neither in equilibrium nor in a condition of constant tidal energy per unit width. They deviate from the ideal ones as result of varying degree of sediment supply, lack of time for adaptation to changing upstream conditions and sea-level rise (Townend, 2012; de Haas et al., 2017). Whether continued*

*sedimentation would reform bar-built estuaries into proper ideal estuaries remains an open question. For our application, the concept of ideal estuaries is useful to assess the degree of deviation from ideal because of the width variations observed as bars, tidal flats and saltmarsh. While we expect a somewhat different degree of convergence such that the ideal state of constant energy per unit of width is approximately maintained, we do not study the deviation of this convergence length from that in ideal estuaries.*

*Ideally, we would want to assess the degree to which an estuary is in equilibrium from an aerial photograph, because this is often the only data available. However, the only indicator derivable from aerial photography is channel width and thus deviation from a converging width profile. Therefore, in Leuven et al. (2017), we defined the excess width, which is the local width of the estuary minus our approximation of the potential ideal estuary width. Here, the ideal estuary width is approximated as an exponential fit on the width of the mouth and the width of the landward river. While the empirical measure of 'ideal width' should not be confused with the 'ideal state' of an estuary, it is the only practical way to estimate deviation from an ideal estuary based on the estuary outline only. Moreover, it proved to be a good indicator of occurring bar patterns (Leuven et al., 2017) and will therefore be applied in this paper to study hypsometries."*

*It should be noted here as well that a linear along-channel depth profile can also be a horizontal bed profile in the case that the predicted channel depth based on hydraulic geometry at the landward side and the predicted depth at the mouth based on tidal prism-CSA relations is equal. We clarified this in the text: "Width-averaged depth profiles along estuaries are often (near-) linear (Savenije, 2015; Leuven et al., 2017), which includes horizontal profiles with constant depth."*

As to the use of hypsometry, it should be better clarified, from the very beginning, that the theoretical framework is quite different from the one proposed for river basins (Strahler, 1952) and for tidal basins (Boon and Byrne, 1981) because in this case the hypsometric curve is applied across channel width (it is a cross-sectional hypsometric curve). I find the idea clever and interesting, but I'd like to see some more discussion on the reasons leading the authors to set up such an analysis. In addition, in the case of the river and tidal basin, the morphological evolution was accounted for, suggesting that different shapes of the hypsometric curve were associated to young or old systems. Is there any possibility of making such an analogy within the framework proposed by the authors? Can the framework account for the dynamic nature of estuarine landscapes? I also wonder if the framework could be applied to any type of estuary of if there are some limitations. Can micro- and macrotidal systems behave in a different way?

*We now clarified that our approach is different from classical hypsometry studies in the first paragraph of the section about the general hypsometric curve: "(…) While it is of interest to use these empirical relations to predict the occurring altitude variation of a landform, the framework here is different, because in this case the hypsometric curve is applied across channel width: it is a cross-sectional hypsometric curve that we change along the system. We aim to use the general hypsometric curve to characterise the occurring cross-sectional hypsometry, (…)".*

*We added a paragraph in the discussion about need for a physically based determination of the hypsometry, and the lack of theory available to do so, also suggested by reviewer 1. A more physics based theory for hypsometry would be required to answer the open questions proposed by the reviewer.*

Finally, I remembered of a paper proposing quite a similar analysis (Toffolon and Crosato, JCR 2017). I think the paper would benefit from recalling the results of the above paper (analyses were applied to the Scheldt estuary). In that paper, the authors analyzed the case of U-shaped, V-shaped, Y-shaped cross section. This could be done also within this framework, to predict the tendency of the estuary to develop particular shapes.

*We were aware of this paper but showed in our earlier work that their application of bar theory is flawed. Indeed, they used a similar fitting approach as we did, but (1) used a power function instead of the Strahler formulation and (2) fitted hypsometric profiles for 15 zones along the estuary and characterise their shape. The effect is that they completely smooth out all differences between bars and channel zones which is precisely the point of our paper. Their U-shaped profiles correspond to our concave profiles, Y-shaped to our convex and V-shaped is exactly intermediate. We added references to their methodology in the introduction and methods section of our paper and compare the results of both studies in the discussion.*

Introduction: *"Furthermore, hypsometry was used as a data reduction method **to characterise entire reaches spanning bars and channels in estuaries (Toffolon & Crosato, 2007} and** shapes of individual tidal bar tops (de Vet et al., 2017)."*

Methods: *"While it is of interest to use these empirical relations to predict the occurring altitude variation of a landform, the framework here is different, because in this case the hypsometric curve is applied across channel width: it is a cross-sectional hypsometric curve that we change along the system. We aim to use the general hypsometric curve to characterise the occurring cross-sectional hypsometry, which is similar to the approach of Toffolon & Crosato (2007) who fitted a power function to 15 zones along the Western Scheldt. However, the zoned approach smooths out all differences between bar complex and channel-dominated zones, which are of interest for this study."*

Discussion: *"These findings are consistent with hypsometry zonations previously found for the Western Scheldt with more concave hypsometries in for channel-dominated morphology and more convex hypsometries for bar complex morphology (Toffolon & Crosato, 2007). Our cross-sectional approach additionally revealed quasi-periodic behaviour within these zones."*

**Detailed comments**
*Detailed textual comments were mostly incorporated and sometimes used as indicator where text clarity had to be improved. We attach a PDF that highlights the changes to the first submission.*

Page 1, Line 19 change "hydrodynamical" to "hydrodynamic"
*Corrected.*

Page 1, Line 22. It should be clarified that the loss of tidal energy by friction is balanced by the gain in tidal energy by convergence.
*Clarified.*

Page 4, Line 5. "width" should be "with"
*Corrected.*

Page 5, Lines 4-5. Actually, this could be the other way round: the presence of bars generates excess width.
*Added to the sentence.*

Page 6, line 1 and line 5. I do not think there is the need to recall that "e" is Euler's number (actually, e^(*-x/Lw) is an exponential function) and "ln" is the natural logarithm.
*Removed.*

Page 6, line 8. Computation of channel width at the landward limit is unclear. Please explain.
*We added: "… and the landward width was measured between the vegetated banks."*

Page 7, lines 16-20. These lines should be rephrased. If I understood correctly, in the first case, fitting is performed on both r and z (as in the third case with the inverted function).
*We added "for r and z" in this sentence: "First, a regular least-squares curve fitting was used for r and z …", before explaining that the results were along-channel varying for z and rather constant for r.*

Page 7, lines 25-27. Please discuss why the inverted function was used.

*We clarified that the original function doet not fit well for cases where we observe steep transitions ffrom the bar top to channel bottom: "To do so, the original formulation of Strahler (1952) was inverted to allow for hypsometries that describe steep transitions from bar top to channel bottom, because the original does not nearly fit as well"*

Page 9, lines 5-7. The linear decrease in water depth from the mouth to the landward section is an assumption that needs be discussed (also in view of other theoretical frameworks developed for tidal channels, e.g. Toffolon and Lanzoni, JGR 2010). In addition, is such an assumption consistent with those embedded in the "ideal estuary" model?

*This comments relates to the first general comment of the reviewer as well as a comment of reviewer 1. Toffolon and Lanzoni (2010) report a constant-depth channel to form when convergence is strong, which is in agreement with the definition of an ideal estuary. We now added a section to the methods in which we clarify the definition of an ideal estuary, together with a description of how we derived a geometric property that characterises deviation from an ideal shape. We now cite this work where we give the definition.*

Page 9, equation (5). Please note that computing the tidal prism by multiplying estuary surface area by the tidal range tantamount to assume a flat water surface elevation along the estuary and moreover does not account for the fact that portions of the estuary area A(t) might get dry during the tidal cycle (see Boon, 1975).

*We added the two assumptions to the text here.*

Figure 3. This figure should be modified. In my view it is a bit confusing to use the same axes for the two columns of panels. The left column should have plots with "bed elevation" on the vertical axis, while the right one should have h_z.

*Done. The right column y-label now reads: "$h_z$, normalised bed elevation (-)".*

Page 10, Figure 5. How was the typical profile for both cases obtained? Please clarify.

*Clarified. Sentence now reads: "Cross-sectional profiles were extracted along the centreline (Section 2.1) and were subsequently classified …"*

Page 10, Caption of Figure 5. "disected" should be "dissected".

*Corrected.*

Page 10, line 6. "suggest" should be "suggests".

*Corrected.*

Page 12, line 7. "In general, the width at the mouth of the estuary and at the upstream estuary is close to ideal ..." shouldn't this be straightforward, due to the fact that you impose those BCs in eq. (2) to compute the ideal along-channel width? Please clarify.

*This is indeed as expected, but not straightforward when there are bars present at the chosen boundary location. We clarified this in the text now (bold parts added): "In general, the width at the mouth of the estuary and at the upstream estuary is close to ideal **by definition** and the hypsometry is concave, **except in systems with wide mouths and bars in the inlet**."*

Page 12, lines 26-30. The reader might wonder why the predictor equation was not applied to the other two estuaries analysed in the manuscript.

*The predictor was applied to all four systems (Figure 11). Nevertheless, Figure 10 only shows the along-channel variability for two systems, which was what we meant to say. We clarified this in the text and refer to both Figure 10 and 11 now.*

Page 15 line 5. I find it difficult to support and discuss the results by citing papers that are still in review or in preparation. Please remove, provide other references or update.
*Removed reference to Kleinhans et al. (2018) and added Leuven et al. (2018) as a preprint to ESSOAr.*

[revised manuscript text omitted]